# Noise Is Not the Main Factor Behind the Gap Between Sgd and Adam on Transformers, but Sign Descent Might Be

**Frederik Kunstner, Jacques Chen, J. Wilder Lavington & Mark Schmidt**[†]
University of British Columbia, Canada CIFAR AI Chair (Amii)[†]
`{kunstner,jola2372,schmidtm}@cs.ubc.ca`
`jacquesc@students.cs.ubc.ca`

## Abstract

The success of the Adam optimizer on a wide array of architectures has made it the default in settings where stochastic gradient descent (SGD) performs poorly. However, our theoretical understanding of this discrepancy is lagging, preventing the development of significant improvements on either algorithm. Recent work advances the hypothesis that Adam and other heuristics like gradient clipping outperform SGD on language tasks because the distribution of the error induced by sampling has heavy tails. This suggests that Adam outperform SGD because it uses a more robust gradient estimate. We evaluate this hypothesis by varying the batch size, up to the entire dataset, to control for stochasticity. We present evidence that stochasticity and heavy-tailed noise are not major factors in the performance gap between SGD and Adam. Rather, Adam performs better as the batch size increases, while SGD is less effective at taking advantage of the reduction in noise. This raises the question as to why Adam outperforms SGD in the full-batch setting. Through further investigation of simpler variants of SGD, we find that the behavior of Adam with large batches is similar to sign descent with momentum.

## 1 Introduction

Adam (Kingma and Ba, 2015) and its derivatives have been so successful in training deep learning models that they have become the default optimizer for some architectures. Adam often outperforms stochastic gradient descent (SGD) by such a margin that SGD is considered incapable of training certain models, to the point of being omitted from performance comparisons (e.g. Liu et al., 2020; Anil et al., 2019). Despite this success, we still do not understand why Adam works, much less why it can outperform SGD by such a wide margin. We have made progress understanding why it should not, as in the work of Reddi et al. (2018) who pointed out that Adam does not converge even on convex problems, but this does not answer why Adam outperforms SGD.

**The limited effectiveness of standard theory.** We usually analyse optimization algorithms under assumptions like Lipschitz continuous function/gradient and convexity (e.g. Nesterov, 2018, Chapters 2–3). Many works have focused on improving the analysis of Adam and its variants under those same assumptions. But these assumptions are only models of how losses behave. They do not convey the complexity of the optimization process in complex architectures, and are limited to showing that Adam does not do much worse than gradient descent (Défossez et al., 2022; Alacaoglu et al., 2020). Analyses in online learning also struggle to illuminate the gap. The assumption that the gradients come from an adversary requires decreasing step-sizes (e.g. Hazan, 2022, Thm 3.1), which decrease too quickly to perform well in practice. Our theoretical understanding is thus still limited in that we cannot describe the empirical behavior we observe—that Adam outperforms SGD in many settings.

As a result, there is a sentiment in the community that the success of these heuristics need not be due to robust theoretical underpinnings, but rather to social dynamics and a co-evolution of deep learning architectures and optimization heuristics (see for example Orabona, 2020). These "adaptive" algorithms might actually be adapted to type of problems where they outperform SGD. But this suggests that they are leveraging some problem structure that our current theory and theory-derived algorithms are missing. Understanding this structure may be key to develop better practical algorithms.

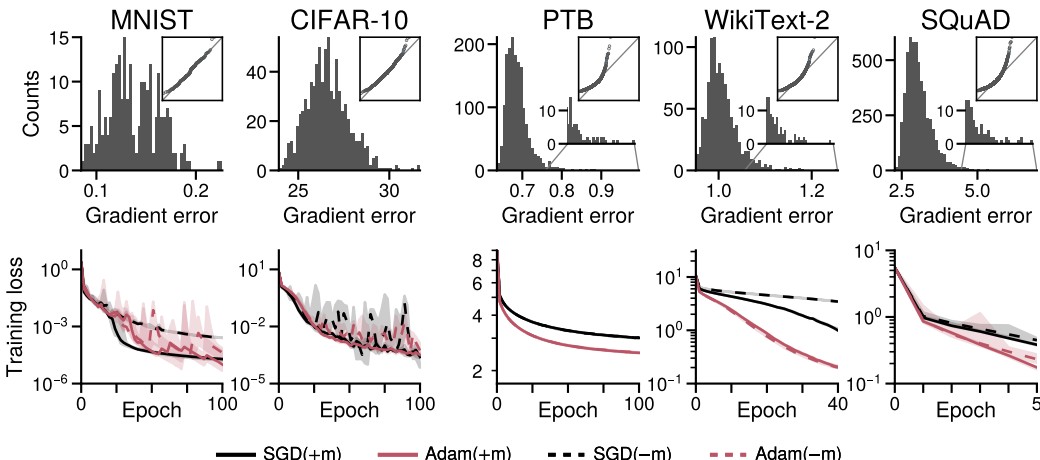

Figure 1: **The Heavy-Tail hypothesis: the gap between SGD and Adam is caused by a heavier tail in the distribution of the stochastic gradient error.** The performance gap between SGD and Adam is larger and more consistent on transformers on text data (right: PTB, Wikitext2, SQuAD) than on CNNs on image data (left: MNIST, CIFAR-10), which coincides with a heavier tail in the distribution of the stochastic gradient error. J. Zhang et al. (2020b) hypothesize that heavier tails might be the cause of this gap. **Top:** Distribution of errors in stochastic gradients at initialization ($\|g - \tilde{g}\|$ where $\tilde{g}$ is stochastic and $g$ is a full gradient) compared against a Gaussian (QQ-plot). **Bottom:** SGD and Adam with and without momentum ($+$m/$-$m) with small batch sizes.

**The heavy-tailed assumption.** Recent works have proposed alternative assumptions to model the behavior of optimizers on neural networks. One such assumption comes from J. Zhang et al. (2020b), who hypothesize that the gap in performance might arise from a *heavy-tailed distribution* of the error induced by stochasticity. They notice a larger and more consistent gap in the performance of SGD and Adam on language models than on image models, which coincides with a heavier tail in the distribution of the stochastic gradient error.[1] We reproduce their observation in Figure 1. The proposed mechanism is that heavy-tailed errors have a larger impact on SGD than on Adam or gradient clipping, as the introduction of bias in the estimation of the gradient reduces its variance. This hypothesis suggests that a path to designing better algorithms is to improve robustness to heavy-tailed noise. For example, Gorbunov et al. (2020) combine acceleration and gradient clipping, while Srinivasan et al. (2021) leverage estimators tailored to heavy-tailed noise.

**Inconsistency with large batch results.** J. Zhang et al. (2020b) note a correlation between heavy-tailedness and cases where Adam outperforms SGD, and give a mechanism to link heavy-tailed errors to this gap. However, the type of noise is not the only difference between image and language tasks. There is limited empirical evidence that stochasticity is the root cause of this gap. In fact, there are reasons to believe noise might not be a major contributor. For example, the lack of variance in the behavior of the optimizers on language tasks in Figure 1 suggests they are less sensitive to noise than image tasks. Moreover, we would expect the gap to diminish as the noise is reduced by increasing the batch size. However, methods such as LAMB (You et al., 2020) or plain Adam find success in large batch settings (Nado et al., 2021), suggesting a competitive advantage even with reduced noise. Studies of batch size scaling also find that Adam scales better with batch size (G. Zhang et al., 2019).

**Alternative explanations.** These empirical results cast doubt on the idea that robustness to heavy-tailed noise is the primary factor behind the performance improvement of Adam over SGD. Hypotheses based on deterministic properties might provide better descriptions of the cause of this gap — we discuss them in more details in Sections 4 and 5. One such interpretation is the view of Adam as a variant of *sign descent*, which was a motivation of RMSprop (Tieleman and Hinton, 2012), as studied by Balles and Hennig (2018) and Bernstein et al. (2018). However, it remains unclear whether the performance of Adam can be explained by its similarities to a simpler algorithm, or if the additional changes needed to obtain Adam are necessary to obtain good performance.

---

[1] The distribution of errors between stochastic gradients $\tilde{g}$ and full gradients $g$, $\|\tilde{g} - g\|$, is well approximated by a Gaussian for image models but closer to an $\alpha$-stable distribution for language models.

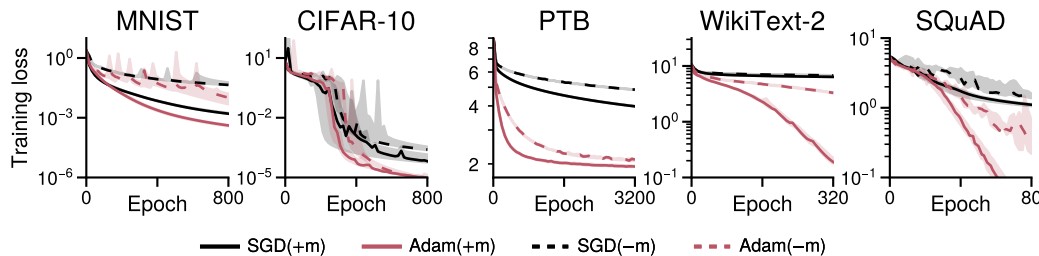

Figure 2: **The gap does not disappear when training in full batch.** Repeating the training procedure of Figure 1 in full batch reveals a similar—or larger—gap between SGD and Adam.

## 1.1 CONTRIBUTIONS

The key question we wish to address is whether the gap between SGD and Adam on transformer models, as studied by J. Zhang et al. (2020b), is really caused by noise, or whether there already is a large difference in the deterministic setting. Our key observations are

1. **Noise is not the key contributor to the performance gap between SGD and Adam on transformers.** Removing noise by running in full batch does not reduce the gap (Figure 2).

2. **SGD benefits less from the reduction in noise than Adam.** Increasing the batch size actually increases the gap between the methods when controlling for the step-size and number of iterations (§3, Figure 3). The performance of SGD does not seem dominated by noise, as its progress per iteration does not improve with batch size as much as it does for Adam (§3.1, Figure 4).

That the performance gap between SGD and Adam grows when noise is removed suggests that the benefit of Adam over SGD can not primarily be due to a robustness to noise. This raises the question as to which component of Adam enables its good performance in the deterministic setting and whether a simpler algorithm can display similar behavior. We show that

3. **Sign descent can close most of the gap between SGD and Adam in full batch.** While sign descent performs poorly with small batch sizes, it improves drastically with larger batch sizes and can bridge most of the gap between gradient descent and Adam (§4, Figure 5–7).

We also present results using normalized gradient updates, which improves on the performance of SGD and scales better to large batch sizes. However, in full batch, sign descent outperforms plain normalization and approaches the performance of Adam. Momentum improves the performance of both Adam and sign descent, and they are similar when both run with- or without momentum.

Despite sign descent being one of the key motivations behind RMSprop, and Adam by extension, the similarity to sign descent in full batch had not been shown empirically. Although this does not give an explanation for why Adam outperforms SGD, it might give an indication of how. As sign descent is a simpler algorithm, understanding its behavior in the deterministic setting might prove a useful tool to understand the success of Adam and the properties of the problem classes where it works well.

## 2 EXPERIMENTAL DESIGN

Our first experiments focus on the performance of SGD and Adam as the batch size increases, and the noise induced by subsampling decreases. Our goal is to check whether the gap between SGD and Adam persists as noise is removed. We touch on the experimental design decisions necessary to interpret the results here. Note that there are practical limitations when comparing optimizers with large batch sizes due to computational challenges, which we discuss in Section 5.1 and Appendix A.

**Problems.** We consider two image and three language tasks.
- Image classification on MNIST and CIFAR-10 using a small CNN and ResNet18.
- Language modeling on PTB and WikiText-2 using a small transformer and Transformer-XL.
- Question answering on SQuAD by fine-tuning a pre-trained DistillBERT model.

Focusing on the heavy-tail hypothesis of J. Zhang et al. (2020b), our attention is on the behavior of the optimizers on language tasks. Image tasks are included for comparison.

**Training performance.** Our goal is to study optimization dynamics in cases where SGD fails to make sufficient progress. Our results focus on the training loss, and we use it to select hyperparameters.

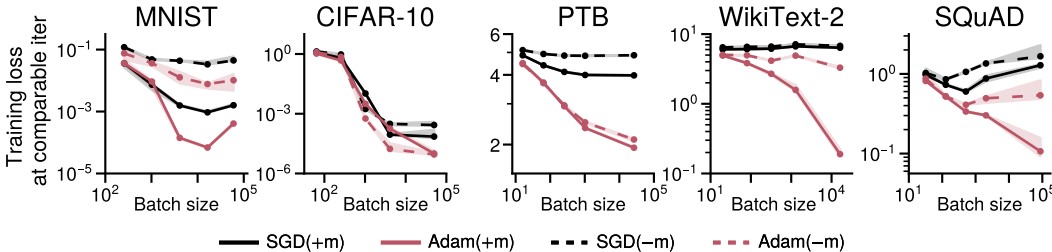

Figure 3: **The gap between SGD and Adam increases with batch size.** Performance after a similar number of iterations across batch sizes. The gap between Adam and SGD grows with batch size on language models, confirming the trend observed in Figures 1 and 2. Due to practical implementation issues, smaller batch sizes still run for more iterations on the larger datasets (WikiText-2, SQuAD) despite being stopped after one epoch (see Appendix A). The degradation in performance as the batch size increases is explained by this decrease in number of iterations. We still observe that the gap grows with batch size despite this bias favoring small batches. To show the effect of batch size beyond the first epoch, we show the full runs in Figure 4.

Results on hold-out data are given in Appendix C, but generalization at large batch sizes is known to require additional tuning and regularization (Shallue et al., 2019; Geiping et al., 2022).

**Batch sizes and reducing noise.** To check if less noisy gradient estimates reduce the gap between SGD and Adam, we measure progress per iteration as we increase the batch size. The batch sizes used, labelled S, M, L and XL, correspond to a $\approx 4\times$ relative increase. Larger batch sizes are run for more epochs, but the increase in batch size leads to fewer iterations (detailed in Tables 1 and 2, Appendix A). For the full batch runs, we remove a small percentage of the data ($\leq 0.5\%$) to ensure the dataset is divided evenly in batches. We check that the observed trends also hold when disabling dropout, the other source of randomness on transformers, in Appendix B.4.

**Simple training procedure.** We use a constant step-size tuned by grid search. While better performance can be obtained with step-size schedules, our primary objective is to gain insight on the behavior of simple updates rather than reproduce state-of-the-art performance.

**Hyperparameter tuning.** Increasing the batch size can allow for larger step-sizes to be stable, and previous work has reported that the best step-size across batch size settings need not be monotonic (Shallue et al., 2019). To account for these effects, we tune the step-size for each batch size individually by grid search. We start with a sparse grid and increase the density around the best step-size. We repeat each run for 3 random seeds. Grid-search validation results are given in Appendix C. We use each method with and without momentum, indicated by (+m) and (−m). When using momentum, we fix it to 0.9 ($\beta$ for SGD, $\beta_1$ for Adam). Other parameters are set to defaults ($\beta_2 = 0.999, \epsilon = 10^{-8}$).

## 3 BEHAVIOR OF SGD AND ADAM IN LARGE BATCH

Comparing SGD and Adam in small and full batch in Figures 1 and 2, we observe the following.

3.a **Adam outperforms SGD by a large margin on language tasks,** as expected from previous observations (J. Zhang et al., 2020b).

3.b **Despite removing stochasticity, the gap between SGD and Adam is not reduced.** It even increases on some problems, especially on the language tasks. On CIFAR-10 and PTB, the performance of Adam improves while the performance of SGD worsens on PTB and WikiText-2.

The importance of momentum is also increased in full batch, as the gap between the same optimizer with and without momentum is larger in Figure 2. But Adam without momentum still outperforms SGD with momentum in large batch on the language tasks.

However, it is not clear that this comparison of the small and full batch settings is a good indicator of the performance *per iteration*. Although run for more epochs, the optimizers take far fewer steps in the full batch setting. To account for this confounding factor, in Figure 3 we introduce intermediate batch sizes and show the loss in each setting when stopped at a comparable number of iterations. The observations from Figures 1 and 2 carry over to Figure 3, which shows the trend across batch sizes more clearly; the gap between Adam and SGD grows with batch size, especially on language models.

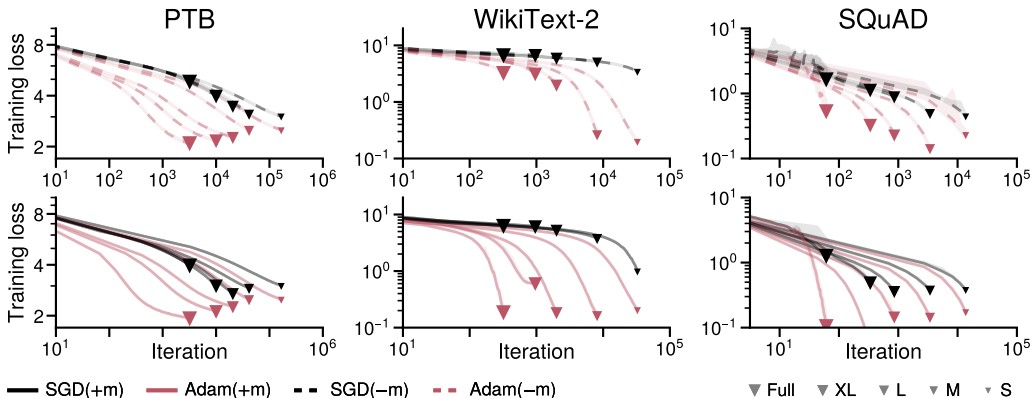

Figure 4: **Adam better takes advantage of the reduced noise with larger batch sizes.** For each problem, each optimizer is shown with five different batch sizes, interpolating between the small batch setting of Figure 1 and the full batch setting of Figure 2. Larger batch sizes run for fewer iterations and terminate earlier, indicated by the markers (▼), with smaller sizes for smaller batches. SGD follow a similar trajectory across most batch sizes, indicating that increasing the batch size past some threshold does not improve the performance; similar results would be expected from running with the same batch size but terminating earlier. Adam, on the other hand, achieves similar error in fewer iterations when the batch size is increased. **Top/bottom:** results without/with momentum.

### 3.1 LIMITATIONS AND ALTERNATIVE VISUALIZATIONS

The view provided by Figure 3 highlights the gap in performance per iteration between Adam and SGD, but does not provide a full picture of their behavior with increasing batch-size. To stop each method at a comparable number of iterations across batch sizes, we have to stop the smallest batch sizes after one epoch. The results thus need not be representative of standard training, as neither SGD nor Adam can achieve reasonable training error after only one epoch.

To verify that the observed behavior holds beyond the first epoch in small batches, we show the full trajectory of the loss for each batch size in Figure 4, focusing on the transformer problems. For each problem, each optimizer is shown running with fives batch sizes, giving a more detailed view of how the gap between the two optimizers increases with batch size, and is greatest at full batch;

3.c **The performance of plain SGD does not significantly improve improves with batch-size.** The trajectory of the loss for SGD is similar across batch sizes, indicating that the reduction in noise from increasing the batch size does not improve its performance; we might expected similar results from running with the same batch size, but terminating earlier.

3.d **Adam improves with batch size and achieves similar or lower error in fewer iterations.**

We also observe that, for both optimizers, momentum improves the convergence speed more with larger batch size. These results are consistent with previous studies on the batch size scaling of momentum (Shallue et al., 2019; Smith et al., 2020) and Adam (G. Zhang et al., 2019).

Our experiments provide additional evidence on language tasks that SGD does not benefit from a reduction in noise in the gradient estimate once beyond a certain batch-size. This suggests that *stochasticity is not the limiting factor in the performance of SGD for these problems.* That the gap between Adam and SGD grows with batch size is also inconsistent with the idea that the improvement of Adam over SGD results from a more robust estimation of the gradient. Those observations do not rule out the possibility that optimization dynamics for small batch sizes are better explained by heavy-tailed rather than sub-Gaussian noise. However, results with large batch sizes suggest that the benefits of Adam over SGD are deterministic in nature.

## 4 ALTERNATIVE INTERPRETATIONS

As noise cannot explain the gap between (S)GD and Adam in the full batch setting, this raises the question *can the improved performance of Adam in full batch be attributed to a simpler algorithm?* Are the algorithmic differences between Adam and SGD, such as moving averages or bias-correction, improving the performance in the stochastic setting, or are they necessary even in the deterministic

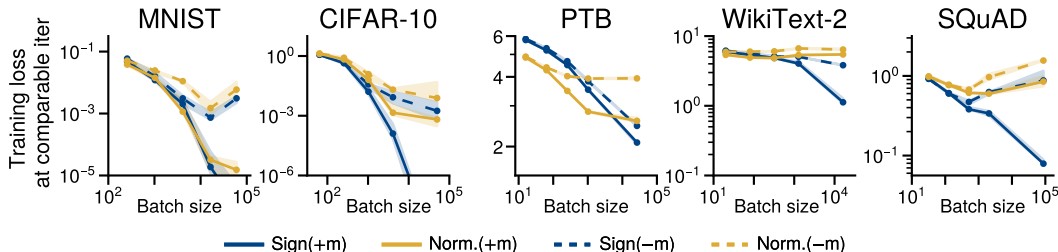

Figure 5: **Sign descent and normalized GD improve with batch size.** Performance after a similar number of iterations across batch sizes. Sign descent performs poorly in small batch sizes and is outperformed by normalized GD. In full batch, sign descent outperforms normalized GD and approaches the performance of Adam. We show the full runs for each batch size in Figure 6.

setting? Our goal is to isolate small algorithmic changes from gradient descent that would exhibit a behavior similar to Adam, while being easier to interpret and analyse. Alternative hypotheses on the performance of Adam might provide insight on its performance, such as its similarity to sign descent or gradient clipping.

### 4.1 SIGN DESCENT AND NORMALIZED UPDATES

A common view of Adam is that it performs a form of smoothed sign descent (Balles and Hennig, 2018; Bernstein et al., 2018). Indeed, the motivation for RMSprop (Tieleman and Hinton, 2012) was to make sign descent more stable in the stochastic setting. The update of RMSprop (with parameters $x_0$, second-moment buffer $v_0 = 0$, and parameters $\alpha, \beta_2, \epsilon$),

$$ v_{t+1} = \beta_2 v_t + (1 - \beta_2)\tilde{g}_t^2 \qquad\qquad x_{t+1} = x_t - \alpha \frac{1}{\sqrt{v_{t+1}} + \epsilon}\tilde{g}_t, $$

reduces to sign descent when setting the hyperparameters $\beta_2$ and $\epsilon$ to 0, as

$$ v_{t+1} = \tilde{g}_t^2, \qquad\qquad x_{t+1} = x_t - \alpha \frac{\tilde{g}_t}{\sqrt{\tilde{g}_t^2}} = x_t - \alpha\,\text{sign}(\tilde{g}_t). $$

The same applies to Adam when taking its momentum parameter $\beta_1$ to 0 (ignoring bias-correction). Consistent with the sign descent interpretation, the success of Adam is often attributed to its normalization effect, that changes across coordinate are uniform despite large differences in their gradients (e.g., Liu et al., 2020, §3.2/Fig. 5). However, why sign descent should work in the deterministic setting and whether RMSprop succeeds in "stabilizing" sign descent have not been explored.

An alternative approach to improve the performance of SGD on language models is to use gradient clipping (Pascanu et al., 2013), Clipping and normalized updates were discussed recently in the context of Adam (J. Zhang et al., 2020a) and sign-based methods (Crawshaw et al., 2022), as element-wise clipping is equivalent to the sign if all elements are larger than the clipping threshold.

To gain insights on whether the success of Adam in the deterministic setting can be attributed to those simpler heuristics, we repeat the prior experiments with the following updates. We use the normalized gradient, which changes the magnitude of the step, and sign descent, which also change its direction. We implement the updates with heavy-ball momentum by accumulating the transformed gradient;

$$ \begin{aligned} m_{t+1} &= \beta m_t + h(g_t), &\text{where} \quad & h(g) = g/\|g\|_2 &\text{for normalized GD}, \\ x_{t+1} &= x_t - \alpha m_{t+1}. & & h(g) = \text{sign}(g) &\text{for sign descent}. \end{aligned} $$

As before, we do not tune momentum and present results with ($\beta = 0.9$) and without ($\beta = 0$) it. We reproduce the visualizations of Figure 3–4 for the new updates in Figure 5–6 and observe that;

4.a **Sign descent performs poorly at small batch sizes but gets closer to Adam in full batch.** At small batch sizes, sign descent can be worse than plain SGD. However, the improvement from increasing batch sizes is greater than for other optimizers, and sign descent goes from being the worst option with small batches to being competitive with Adam in full batch.

4.b **Normalized GD scales better but plateaus.** While normalizaton improves performances and helps scale with batch size, the scaling plateaus earlier than for sign descent.

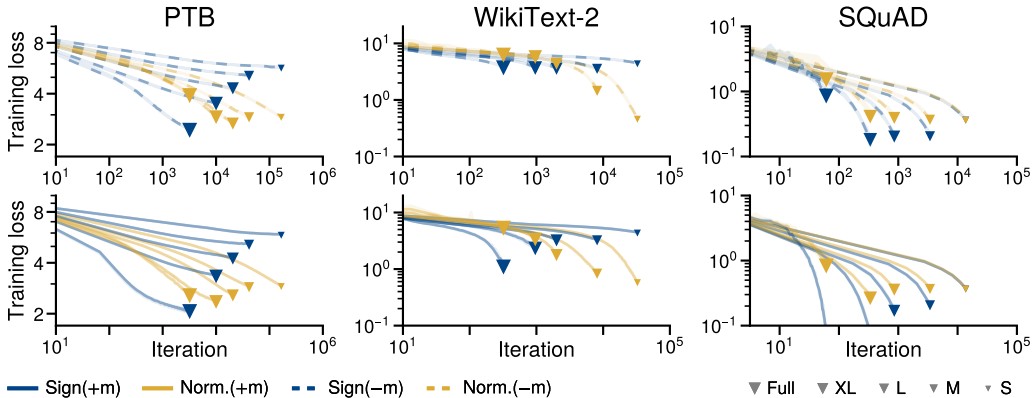

Figure 6: **Sign descent improves with batch size.** While normalized GD outperforms sign descent in small batch sizes, the relationship switches in full batch and sign descent outperforms normalized GD. Normalized gradient descent also scales better with batch size than plain gradient descent (compare Figure 4), but the method that scales best with batch size is sign descent. Larger batch sizes run for fewer iterations and terminate earlier, indicated by the markers (▼), with smaller sizes for smaller batches. **Top/bottom:** results without/with momentum.

As with other optimizers, momentum helps both sign descent and normalized GD scale to larger batch sizes. The improvement in performance with batch size observed for Adam is closer to that of sign descent than normalized GD. In full batch, sign descent with momentum closes most of the gap between SGD and Adam, and it can even outperform Adam on some problems.

To better illustrate that the performance of sign descent is poor in small batch, but that it closes most of the gap between GD and Adam in full batch, we compare all optimizers with momentum in a traditional loss per epoch format in Figure 7 (without momentum in Figure 10, Appendix B).

While a gap still exists between Adam and sign descent, the improvement in performance over gradient descent supports the hypothesis that the performance gap between Adam and gradient descent has its roots in a difference in the deterministic case. Parts of the benefit of the second-moment estimation in Adam or RMSprop can be attributed to the difference between gradient descent and sign descent. The good scaling properties of Adam to large batch sizes are also shared with sign descent. This connection might provide a new avenue for analysis, by studying a simpler algorithm, to understand the behavior of Adam and the impact of noise.

## 5 DISCUSSION

We present experimental evidence that runs counter to the idea that Adam outperforms SGD on transformers due to a more robust estimate of the descent direction. Rather, experimental results show that Adam outperforms gradient descent in the full batch setting, where gradient descent already performs poorly. Those results validate the intuitive motivations for RMSProp—and by extension, Adam—that the sign of the gradient might be a more reliable direction than the gradient. In addition, the results show that the improvement in performance with large batch sizes exhibited by Adam, also observed by previous work (G. Zhang et al., 2019), is shared by sign descent. As Adam outperforms sign descent in small batch sizes, this suggests that the benefits of the algorithmic differences between sign descent and Adam indeed improves its behavior in small batch sizes.

Despite the reported similarities between sign descent and Adam, that sign descent outperforms gradient descent by such a margin in full batch on transformers is surprising. The sign descent perspective does not fit the common view of Adam as an "adaptive" algorithm, and there is limited theory explaining the benefits of sign descent in deep learning. But that sign descent performs well on deep learning tasks is corroborated by the recent work of Chen et al. (2023). Using evolutionary search to discover optimization algorithms, they obtain a variant of sign descent with momentum that outperforms Adam variants on a range of deep learning models and exhibits better relative performance with large batch sizes. Understanding how to capture the benefits of sign descent on

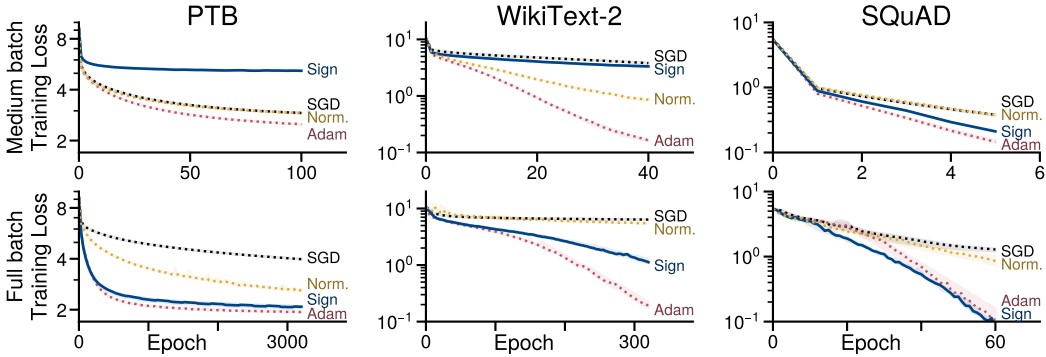

Figure 7: **Sign descent can close most of the gap between GD and Adam in full batch.** At small batch sizes, the performance of sign descent can be worse than SGD. However, the improvement when increasing the batch size is greater than for other optimizers, and sign descent goes from being the worst option to being competitive with Adam in full batch. All optimizers are shown with momentum. A similar trend holds when looking at all optimizers without momentum, shown in Figure 10. **Top/Bottom:** performance with a medium batch size/in full batch.

those models might give us a better understanding of the loss landscape in deep learning. We end by discussing potential open questions, related work, and limitations of the results presented here.

**Improvement of sign descent with batch size.** Larger improvements with batch size are often associated with methods that are faster than gradient descent, such as second-order methods, momentum or Nesterov acceleration. This improved performance typically comes at the cost of an increased sensitivity to noise, requiring larger batch sizes to achieve better performance. This has been shown empirically for momentum on deep learning models (Shallue et al., 2019; Smith et al., 2020), and can be established formally, e.g. on quadratics (Flammarion and Bach, 2015). Quadratic models have also been used to justify the improved performance of Adam with large batches (G. Zhang et al., 2019). This justification relies on the interpretation of Adam as an approximate second-order method, similar to the KFAC approximation to natural gradients (Martens and Grosse, 2015). However, this interpretation of Adam seems incompatible with its interpretation as a form of sign descent, and the quadratic model alone seems insufficient to explain the performance of sign descent.

**Limitations of typical assumptions.** Typical analyses in optimization assume the objective function is smooth (has bounded Hessian, $\|\nabla^2 f\| \leq L$) to ensure the accuracy of its Taylor approximation. This assumption implies the following, sometimes called the descent lemma (Bertsekas, 1999, Prop. A.24)

$$f(y) \leq f(x) - \frac{1}{2L}\|\nabla f(x)\|^2 + \frac{L}{2}\|y - (x - (1/L)\nabla f(x))\|^2.$$

Without further assumptions, this bound is the only guarantee of performance. As the bound is optimized at the gradient descent step $y = x - (1/L)\nabla f(x)$, one would expect any corollary to not improve on GD. It is of course possible to provably improve on GD with further assumptions. e.g. with acceleration on convex functions (Nesterov, 1983), but this highlights the strong relationship between the typical smoothness assumption and GD. Moreover, recent works have called into question the adequacy of this assumption, showing that it does not hold even in "simple" deep learning problems (Cohen et al., 2021). We discuss next recent works that explore relaxations of smoothness to better characterize the loss landscape and capture the benefit of clipping and sign-based algorithms.

**Justifications for normalization approaches.** J. Zhang et al. (2020a) provides empirical evidence that, on deep learning models, the Hessian and gradient norm can share a similar scale. This observation motivates a relaxation of the uniform smoothness bound ($\|\nabla^2 f\| \leq L$) to allow the Hessian to grow with the gradient ($\|\nabla^2 f\| \leq L_0 + L_1\|\nabla f\|$). This suggests $f$ behaves like a smooth function when the gradient is small, and a gradient descent step can be expected to make progress. But if the gradient is large, the Hessian might be large. Dividing by the upper bound on the Hessian, the gradient norm, normalizes the step and leads to clipping. This view of clipping has been further explored by B. Zhang et al. (2020) and Qian et al. (2021). However, our experiments suggest that

normalization alone might be not be sufficient to bridge the gap between SGD and Adam, and that element-wise clipping or sign-based methods exploit additional structure about the problem.

**Justifications for sign- and element-wise clipping approaches.** Bernstein et al. (2018) and Balles et al. (2020) provide conditions under which sign-based approaches can outperform SGD in the deterministic or stochastic setting, assuming element-wise smoothness or measuring smoothness in the $L_\infty$ norm. As with the standard smoothness assumption, it is unclear whether they provide an accurate description of the optimization dynamics in deep learning. Following the relaxed smoothness assumption of J. Zhang et al. (2020a), Crawshaw et al. (2022) propose a coordinate-wise relaxation to analyse a variant of sign-descent inspired by Adam, using exponential moving averages to estimate the sign. While providing a justification for sign descent-like algorithms, it is not clear whether the coordinate-wise relaxation is practical. Having $2d$ parameters, one for each of the $d$ weights in the network, checking this variant requires checking each weight and convergence results depend on the $2d$ problem specific-constants. Identying a verifiable assumption that captures the advantage of sign-based methods in the deterministic setting and understanding when it holds remains open.

**What architecture choices lead to the increased gap between SGD and Adam?** We do not observe such a large gap between SGD and Adam on CNNs on image tasks compared to transformers on language tasks. What properties of the problem or architecture leads to a much better performance of Adam, sign descent or other normalization schemes, beyond the difference in the tails of the stochastic gradients noted by J. Zhang et al. (2020b)? The prediction over a large vocabulary in language tasks—much larger than the number of classes in image tasks—could suggest a similarity with overparametrized logistic regression, which is known to exhibit faster convergence of normalized gradient descent (Nacson et al., 2019). The architectures also differ in their choice of normalization layers, skip connections and initializations, choices which were crucial for the training of deep networks on image tasks (He et al., 2016) but might not yet be as mature for transformer models. Reducing the complexity of such model to a minimal example that still exhibits a large gap between SGD and Adam but without the computational hurdles of large deep learning models would be helpful to serve as a mental model and benchmark to test hypotheses or optimizers.

**Designing versions of sign descent that are robust to small batch sizes?** The exponential moving average used by RMSProp and Adam works well in practice, but capturing this benefit in theory is challenging. Existing guarantees show a negative dependence on $\beta_2$ (Défossez et al., 2022; Alacaoglu et al., 2020; Crawshaw et al., 2022). In contrast, alternative mechanisms such as majority voting, popular in the distributed setting due to the low communication cost of sign descent, have provable guarantees (Bernstein et al., 2018; Safaryan and Richtárik, 2021). How to obtain similar guarantees for exponential moving averages, which are known to work for deep learning, remains open.

## 5.1 LIMITATIONS

**A major limitation whenever probing the full batch setting is computational costs.** As we measure progress per iteration, keeping the number of iterations constant across batch sizes would be ideal, but it is computationally infeasible given our budget. We mitigate this limitation by introducing intermediate batch sizes that interpolate between the small and full batch setting. However, due to those limitations, our results might only probe the "beginning" of the training procedure compared to state-of-the-art workloads. We also use a constant step-size, and some of the difficulties encountered by SGD on transformers might be mitigated by a warm-up or decaying step-size schedule if the initialization has a strikingly different landscape from the remainder of the optimization path.

**Focus on training performance.** We do not attempt to optimize hyperparameters for validation error, and instead focus on understanding the optimization performance on the training loss. The results might be of limited applicability in practical settings focused on generalization. However, the validation metrics (available alongside our grid-search results in Appendix C) suggests that optimizers which perform better on training error achieve a lower testing loss faster (before over-fitting).

**Qualitative differences across architectures.** The striking difference in performance between SGD and Adam on transformers need not hold for other architectures. On the two image tasks included to serve as comparison points, the distinction between SGD and Adam and the observed trends are less clear. For example, on ResNet18/CIFAR10, SGD behaves differently than on the language tasks in Figure 8 (Appendix B), and improves with batch size as much as Adam. We hypothesize this is due to the network using Batch Normalization. We might need different assumptions, analysis tools and possibly optimizers for different families of architectures that have qualitatively different behaviors.

ACKNOWLEDGMENTS

We thank Si Yi (Cathy) Meng, Aaron Mishkin, and Victor Sanches Portella for providing comments on the manuscript and earlier versions of this work. This research was partially supported by the Canada CIFAR AI Chair Program, the Natural Sciences and Engineering Research Council of Canada (NSERC) Discovery Grants RGPIN-2022-03669, and enabled in part by support provided by the Digital Research Alliance of Canada (`alliancecan.ca`).

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

# Supplementary material

## A  EXPERIMENTAL DETAILS

The code to reproduce our experiments and data generated by each run is available at https://github.com/fkunstner/noise-sgd-adam-sign

### A.1  DATASETS AND MODELS

Problems in the main text are referred to by the dataset and corresponds to the following problems. For all problems, we use the default train/test split and report the performance on the test split as validation error.

MNIST, Modified LeNet5                                                                 (LeCun et al., 1998)

> Image classification task using a modified LeNet5 architecture; a 3-convolution, 2-linear layer average pooling and tanh activations, using a linear rather than Gaussian output.
>
> Dataset   yann.lecun.com/exdb/mnist/

CIFAR10, ResNet18                                                          (Krizhevsky, 2012; He et al., 2016)

> PyTorch (Paszke et al., 2019) implementation of the 18-layer CNN with residual connections.
>
> Dataset   www.cs.toronto.edu/~kriz/cifar.html
> Model     pytorch.org/vision/0.10/models.html#torchvision.models.resnet18

PTB, Transformer                                                      (Marcus et al., 1993; Vaswani et al., 2017)

> Word-level language modeling with sequences of 35 tokens using a simple Transformer model used as a tutorial example in PyTorch. The architecture consists of an 200-dimensional embedding layer, 2 transformer layers (2-head self attention, layer normalization, linear(200x200)-ReLU-linear(200x200), layer normalization) followed by a linear layer. Data processing uses the implementation of Dai et al. (2019).
>
> Dataset      catalog.ldc.upenn.edu/LDC99T42
> Processing   github.com/kimiyoung/Transformer-XL
> Model        pytorch.org/tutorials/beginner/transformer_tutorial.html

WikiText-2, Transformer-XL                                              (Merity et al., 2017; Dai et al., 2019)

> Word-level language modeling with sequences of length 128 with Transformer-XL, using the implementation of Dai et al. (2019) for the model and data preprocessing. Hyperparameters follow the ENWIK8 base experiment of Dai et al. (2019), except with the modifications of J. Zhang et al. (2020b), using 6 layers and a target length of 128.
>
> Dataset   blog.salesforceairesearch.com/the-wikitext-long-term-
>           dependency-language-modeling-dataset/
> Model     github.com/kimiyoung/Transformer-XL

SQuAD v1.1, DistillBERT finetuning                                     (Rajpurkar et al., 2016; Sanh et al., 2019)

> Question-answering by fine-tuning a pretrained DistillBERT model using the HuggingFace (Wolf et al., 2020) implementation of the model and data processing.
>
> Dataset   rajpurkar.github.io/SQuAD-explorer/explore/1.1/dev/
> Model     huggingface.co/transformers/v4.9.2/model_doc/distilbert.html

Table 1: **Batch sizes used in experiments.** The batch size corresponds to the number of input-output samples. For image data, a sample corresponds to an image-label pair. For language models on PTB/WikiText-2, a sample corresponds to pairs of sequences of 35/135 tokens. For DistillBERT on SQuAD, a sample corresponds to a question/answer pair. The full batch runs that use $\geq 99.5\%$ of the entire dataset, drop $0.5\%$ of the data and run in full batch on the remaining data at every epoch.

| Dataset/Model | | Batch size | % data | Epochs | Iter./Epoch | Total Iter. |
|---|---|---|---|---|---|---|
| MNIST | S | 256 | 0.43% | 100 | 234 | 23 400 |
| Modified LeNet5 | M | 1 024 | 1.71% | 100 | 58 | 5 800 |
| | L | 4 069 | 6.78% | 200 | 14 | 2 800 |
| | XL | 16 384 | 27.31% | 400 | 3 | 1 200 |
| | Full | 60 000 | 100.00% | 800 | 1 | 800 |
| CIFAR-10 | S | 64 | 0.13% | 100 | 781 | 78 100 |
| ResNet18 | M | 256 | 0.51% | 100 | 195 | 19 500 |
| | L | 1 024 | 2.05% | 100 | 48 | 4 800 |
| | XL | 4 096 | 8.19% | 200 | 12 | 2 400 |
| | Full | 50 000 | 100.00% | 800 | 1 | 800 |
| PTB | S | 16 | 0.06% | 100 | 1 659 | 165 900 |
| Transformer | M | 64 | 0.24% | 100 | 414 | 41 400 |
| | L | 256 | 0.96% | 200 | 103 | 20 600 |
| | XL | 1 024 | 3.86% | 400 | 25 | 10 000 |
| | Full | 26 520 | 99.85% | 3 200 | 1 | 3 200 |
| WikiText-2 | S | 20 | 0.12% | 40 | 815 | 32 600 |
| Transformer-XL | M | 80 | 0.49% | 40 | 203 | 8 120 |
| | L | 320 | 1.96% | 40 | 50 | 2 000 |
| | XL | 1 280 | 7.84% | 80 | 12 | 960 |
| | Full | 16 240 | 99.53% | 320 | 1 | 320 |
| SQuAD | S | 32 | 0.04% | 5 | 2 741 | 13 705 |
| DistillBERT | M | 128 | 0.15% | 5 | 685 | 3 425 |
| | L | 512 | 0.58% | 5 | 171 | 855 |
| | XL | 2 048 | 2.33% | 10 | 42 | 420 |
| | Full | 87 680 | 99.96% | 60 | 1 | 60 |

## A.2 BATCH SIZES AND EPOCH COUNTS

Our goal is to compare the optimization performance while controlling for the number of iterations. But keeping the iteration count constant across batch sizes is computationally infeasible. For example, running Transformer-XL on WikiText-2 with a batch size of 20 for 40 epoch (32 600 iterations) takes 1–2 hours. Running 320 epochs in full batch (320 iterations) takes 8–10 hours. Running $100\times$ more iterations would take over a month, making hyperparameter tuning infeasible. As increasing the batch size leads to better performance in fewer iterations, we run larger batch sizes for fewer iterations. We control how the number of iterations changes from setting to setting to have comparable results. The batch sizes indicated by S, M, L and XL correspond to a relative increase of $4\times$ in batch size, and tune the number of epochs for each setting so that the number of iterations from one batch size to the next stays within a factor of $\approx$2–8. The settings used in our experiments are described in Table 1.

To highlight the improvement in performance per iteration and account for the confounding factor that larger batch sizes run for fewer iterations (Appendices B.1 and B.2), we introduce intermediate batch sizes and show the loss in each setting when stopped at a comparable number of iterations. The stopping times for these experiments (given in Appendix A.2) were selected to make the number of iterations similar. They could not be made equal as we did not record the loss on the entire dataset at each iteration. For example, on WikiText-2, the full-batch run is stopped after 320 epochs (320 iterations), while the small batch-size run is stopped at one epoch (815 iterations). On the larger language datasets (WikiText-2, SQuAD), the degradation in performance as the batch size increases in Figure 3 is attributable to this decrease in number of iterations. However, we still observe that the gap between SGD and Adam grows with batch size, despite this bias favoring small batch sizes.

Table 2: **Stopping times** used in Figures 3 and 5 to obtain a comparable number of iterations across batch sizes. The number of iterations can not always be made made equal as we did not record the loss on the entire dataset at each iteration, and can only stop the small batch sizes at one epoch, but the number of steps taken by the optimizer are much closer than in Table 1.

| Batch size | | Epoch | Iter. | Batch size | | Epoch | Iter. | Batch size | | Epoch | Iter. |
|---|---|---|---|---|---|---|---|---|---|---|---|
| **MNIST** | | | | **PTB** | | | | **SQuAD** | | | |
| S | 256 | 4 | 936 | S | 16 | 2 | 3 320 | S | 32 | 1 | 2 741 |
| M | 1 024 | 14 | 812 | M | 64 | 8 | 3 320 | M | 128 | 2 | 1 370 |
| L | 4 096 | 58 | 812 | L | 256 | 32 | 3 296 | L | 512 | 3 | 513 |
| XL | 16 384 | 267 | 801 | XL | 1 024 | 128 | 3 200 | XL | 2 048 | 4 | 168 |
| Full | 60 000 | 800 | 800 | Full | 26 520 | 3 200 | 3 200 | Full | 87 680 | 60 | 60 |
| **CIFAR-10** | | | | **WikiText-2** | | | | | | | |
| S | 64 | 1 | 781 | S | 20 | 1 | 815 | | | | |
| M | 256 | 4 | 780 | M | 80 | 3 | 609 | | | | |
| L | 1 024 | 16 | 768 | L | 320 | 10 | 500 | | | | |
| XL | 4 096 | 64 | 768 | XL | 1 280 | 34 | 408 | | | | |
| Full | 50 000 | 768 | 768 | Full | 16 240 | 320 | 320 | | | | |

For settings where computing the gradient would not fit into memory, we use gradient accumulation to reduce peak memory consumption (computing the gradients of smaller batches and averaging them). We use accumulation for Full batch runs, WikiText-2 ($\geq$ L) and SQuAD ($\geq$ M). Due to the batch normalization layers (Ioffe and Szegedy, 2015) in ResNet18, computing the full-batch gradient would require to pass the entire dataset through the network at once (the normalization makes it so that the average of the gradients of the minibatches is different from the full gradient). We use the average over the minibatch gradients with a batch size of 10 000. Transformer models use layer normalization (Ba et al., 2016) and are unaffected.

The transformer models on PTB and WikiText-2 use dropout by default, meaning that the increase to Full batch reduces noise to a floor but does not remove it entirely. In Appendix B.4, we verify that the observation that sign descent bridges the gap between (S)GD and Adam in the deterministic setting on those two datasets in full batch after removing dropout.

### A.3 ALGORITHMS PSEUDOCODE

We use the PyTorch (Paszke et al., 2019) implementation of SGD and Adam, and implement the normalized variants (sign descent, normalized GD) with heavy-ball momentum following the pseudo-code in Algorithm 1. The pseudo-code of Adam is given in Algorithm 2.

For every algorithm, we tune the main step-size by grid search. The momentum parameters $\beta$ in gradient descent variants and $\beta_1$ in Adam is set either to 0 or 0.9. This is indicated in the main text and figure legends by the suffix (+m) (with momentum) or (−m) (without). We leave the remaining hyperparameters of Adam to their default ($\beta_2 = 0.999, \epsilon = 10^{-8}$).

---

**Algorithm 1** Gradient Descent and variants

**Parameters:** step-size $\alpha$, momentum $\beta$.
**Initialization:** momentum buffer $m_0 = 0$.
**At iteration $t$:** given a stochastic gradient $\tilde{g}_t$,

$$m_t = \beta m_{t-1} + d_t$$
$$x_{t+1} = x_t - \alpha m_t$$

$$\text{with } d_t = \begin{cases} \tilde{g}_t & \text{for Gradient Descent} \\ \tilde{g}_t/\|\tilde{g}_t\|_2 & \text{for Normalized GD} \\ \text{sign}(\tilde{g}_t) & \text{for Sign Descent} \end{cases}$$

---

**Algorithm 2** Adam

**Parameters:** step-size $\alpha$, momentum $\beta_1, \beta_2$.
**Initialization:** moments buffers $m_0, M_0 = 0$.
**At iteration $t$:** given a stochastic gradient $\tilde{g}_t$,

$$m_t = \beta_1 m_{t-1} + (1 - \beta_1)\tilde{g}_t$$
$$M_t = \beta_2 M_{t-1} + (1 - \beta_2)\tilde{g}_t^2$$
$$x_{t+1} = x_t - \alpha \frac{m_t/(1 - \beta_1^t)}{\sqrt{M_t/(1 - \beta_2^t)} + \epsilon}$$

Using the default $\epsilon = 10^{-8}$.

## A.4 HISTOGRAMS IN FIGURE 1

The histograms in Figure 1 show the distribution of the stochastic gradient errors at initialization to confirm the observation of J. Zhang et al. (2020b). J. Zhang et al. also show the pattern is preserved through training. We use the batch sizes

| MNIST | CIFAR-10 | PTB | WikiText-2 | SQuAD |
|-------|----------|-----|------------|-------|
| 256 | 64 | 16 | 16 | 16 |

As ResNet18 uses batch normalization, it does not have a well-defined "full" gradient (the average of minibatch gradients is not the gradient obtained by passing the entire dataset through the network). We use the average over the minibatch gradients. Other models are unaffected.

## A.5 GRID SEARCH

We start with a sparse grid of integer powers of 10 (eg. $[10^{-5}, 10^{-4}, \ldots, 10^0]$). After an initial run to identify a reasonable region, we increase the density to include half-powers (eg. $[10^{-3}, 10^{-2.5}, 10^{-2}, 10^{-1.5}, 10^{-1}]$). The density of the grid is the same for all problems. We run those step-sizes for 3 random seeds determining the data ordering and initialization (except for DistillBERT on SQuAD, which is pretrained).

To select the step-size, we minimize the (maximum over seeds of the) training loss at the end of training. *End-of-training* is the epoch reported in Table 1 for most figures. The exceptions are Figures 3 and 5 which use the stopping times in Table 2.

The final performance as a function of the step-size for the training loss and accuracy/PPL/F1-score (including on holdout data) are given in Appendix C

## B ADDITIONAL PLOTS

**Appendix B.1** Gap vs. Batch size (Figure 4 including image tasks)
**Appendix B.2** Gap vs. Batch size (Version of Figure 6 including image tasks)
**Appendix B.3** Relative performance of sign descent in medium vs. full batch (Version of Figure 7 without momentum)
**Appendix B.4** Verifying the results hold without Dropout

## B.1 GAP VS. BATCH SIZE (FIGURE 4 INCLUDING IMAGE TASKS)

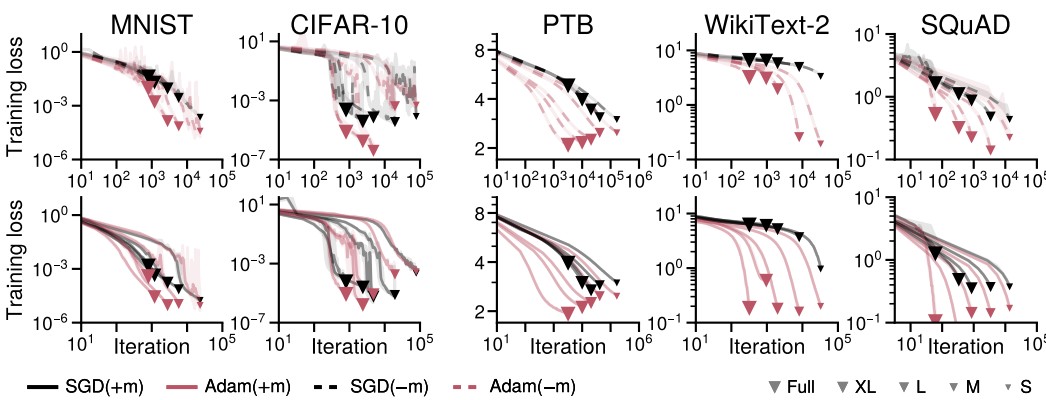

Figure 8: **Adam better takes advantage of the reduced noise with larger batch sizes.** For each problem, each optimizer is shown with five different batch sizes, interpolating between the small batch setting of Figure 1 and the full batch setting of Figure 2. Larger batch sizes run for fewer iterations and terminate earlier, indicated by the markers (▼), with smaller sizes for smaller batches. SGD follow a similar trajectory across most batch sizes, indicating that increasing the batch size past some threshold does not improve the performance; similar results would be expected from running with the same batch size but terminating earlier. Adam, on the other hand, achieves similar error in fewer iterations when the batch size is increased. However, the trends are not as clear on MNIST and CIFAR-10. SGD performs better on CIFAR-10 than on other problems, which might be attributable to the presence of Batch Normalization layers. **Top/bottom:** results without/with momentum.

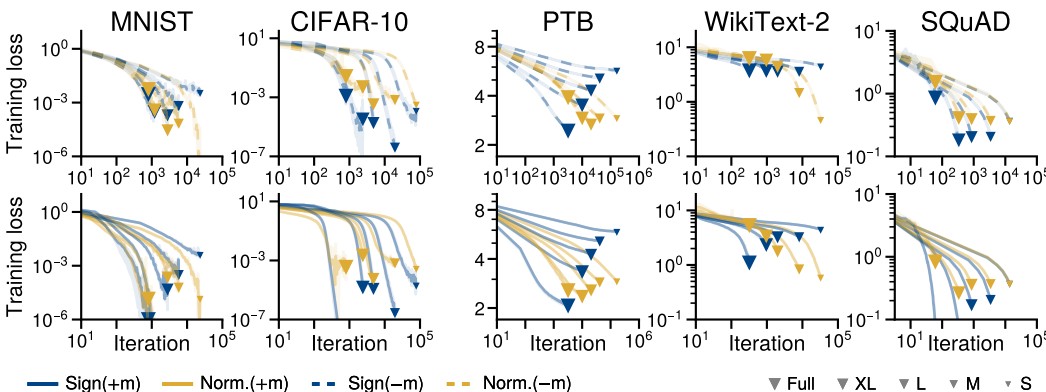

Figure 9: **Sign descent improves with batch size.** While normalized GD outperforms sign descent in small batch sizes, the relationship switches in full batch and sign descent outperforms normalized GD. Normalized gradient descent also scales better with batch size than plain gradient descent (compare Figure 4), but the method that scales best with batch size is sign descent. Larger batch sizes run for fewer iterations and terminate earlier, indicated by the markers (▼), with smaller sizes for smaller batches. **Top/bottom:** results without/with momentum.

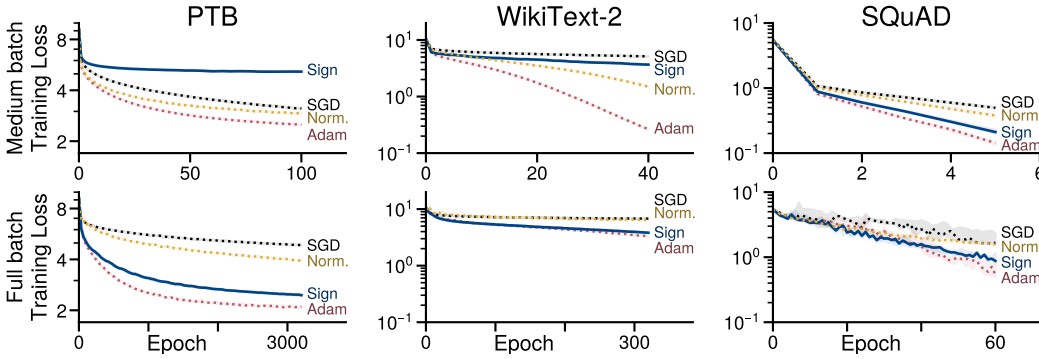

Figure 10: **Sign descent can close most of the gap between GD and Adam in full batch.** At small batch sizes, the performance of sign descent can be worse than SGD. However, the improvement when increasing the batch size is greater than for other optimizers, and sign descent goes from being the worst option to being competitive with Adam in full batch. All optimizers are shown without momentum. Although all optimizers suffer from the lack of momentum, especially on WikiText-2, sign descent closes most of the gap between SGD and Adam. A similar trend holds when looking at all optimizers with momentum, shown in Figure 7. **Top/Bottom:** performance with a medium batch size/in full batch.

### B.4 VERIFYING THE RESULTS HOLD WITHOUT DROPOUT

The increase of batch size to "Full batch" drives noise to a floor, but does not remove it entirely due to the use of Dropout in the transformer models. Figure 11 shows those models after disabling dropout to verify that the same trends holds, showing the performance of the optimizer in full batch, as in the bottom row of Figure 7 (momentum) and Figure 10 (without momentum). The settings use are the same as for the Full batch in those figures (See `Full` in Table 1 for batch size and epoch counts) and uses step-sizes tuned by grid-search, with the only modification of disabling dropout. We note that the models achieve lower training error, but the overall trend of Sign Descent closing most of the gap between SGD and Adam in full batch is preserved.

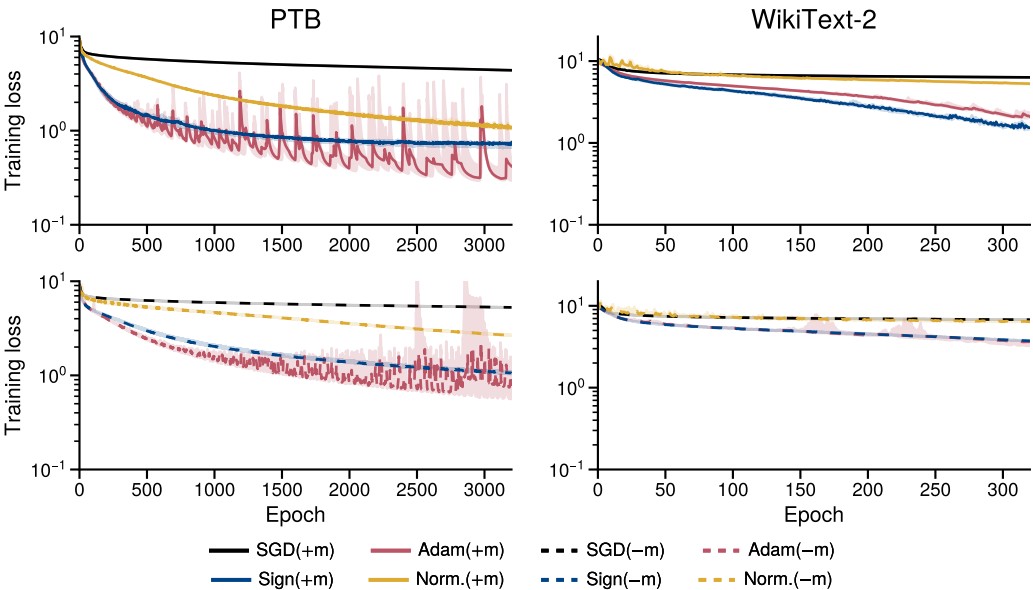

Figure 11: **Checking that sign descent still bridges the gap between GD and Adam when dropout is removed.** The increase of batch size to "Full batch" drives noise to a floor, but does not remove it entirely due to the use of dropout in the transformer models. We rerun those models after disabling dropout to verify that the same trends observed in Figure 6-7 holds. We note that the models achieve lower training error, but the overall trend of Sign Descent closing most of the gap between SGD and Adam in full batch is preserved.

## C  GRID-SEARCHES AND PERFORMANCE OF SELECTED RUNS

### C.1  SGD AND ADAM

#### C.1.1  MODIFIED LeNet5 ON MNIST

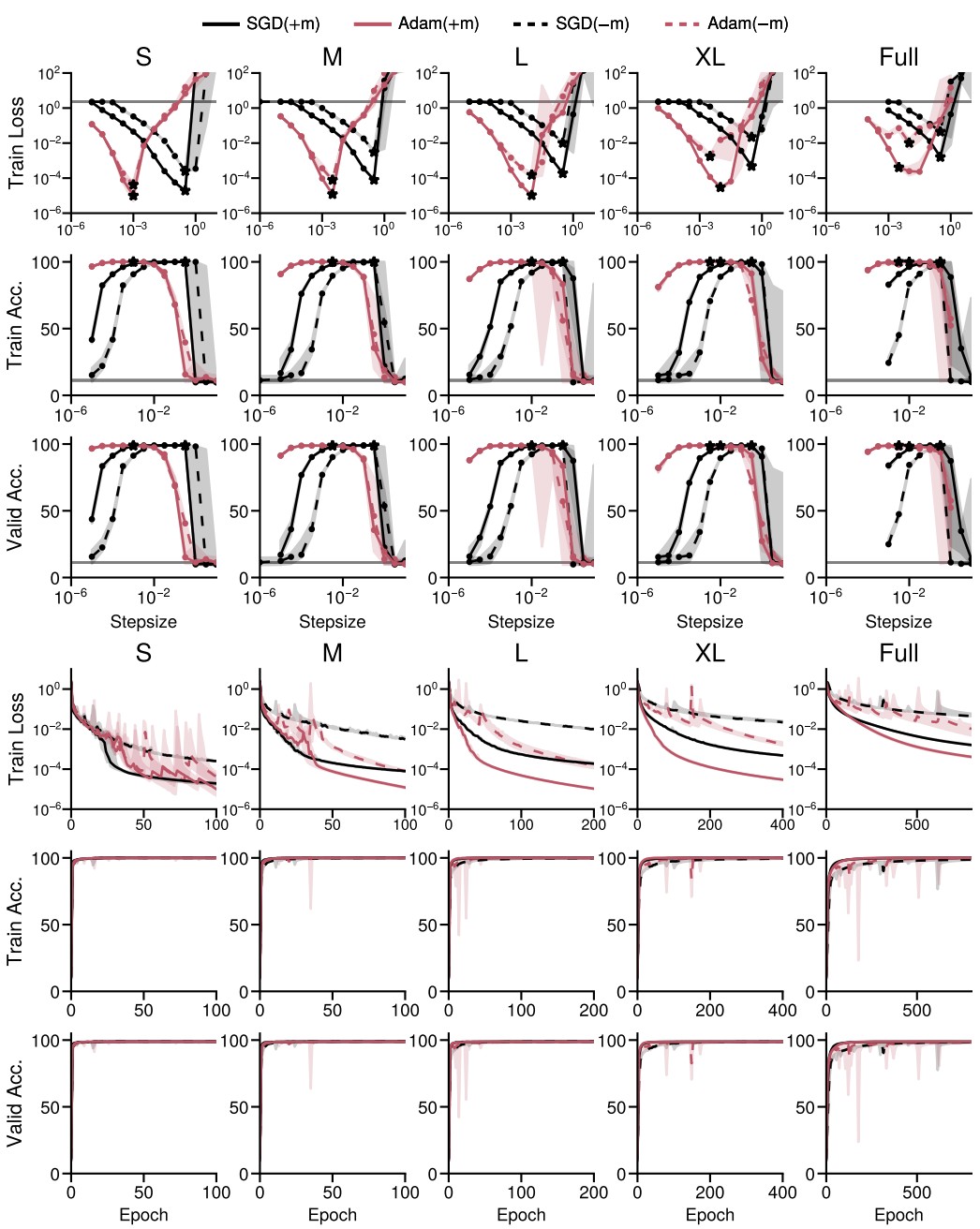

Figure 12: **Grid search and selected runs for each batch size modified LeNet5 on MNIST**. **Top:** Loss and evaluation metrics at the end of training (see Table 1) for step-sizes evaluated. Grey line corresponds to value at initialization. **Bottom:** Performance of the selected step-size on training loss and evaluation metrics during training.

### C.1.2  RESNET18 ON CIFAR10

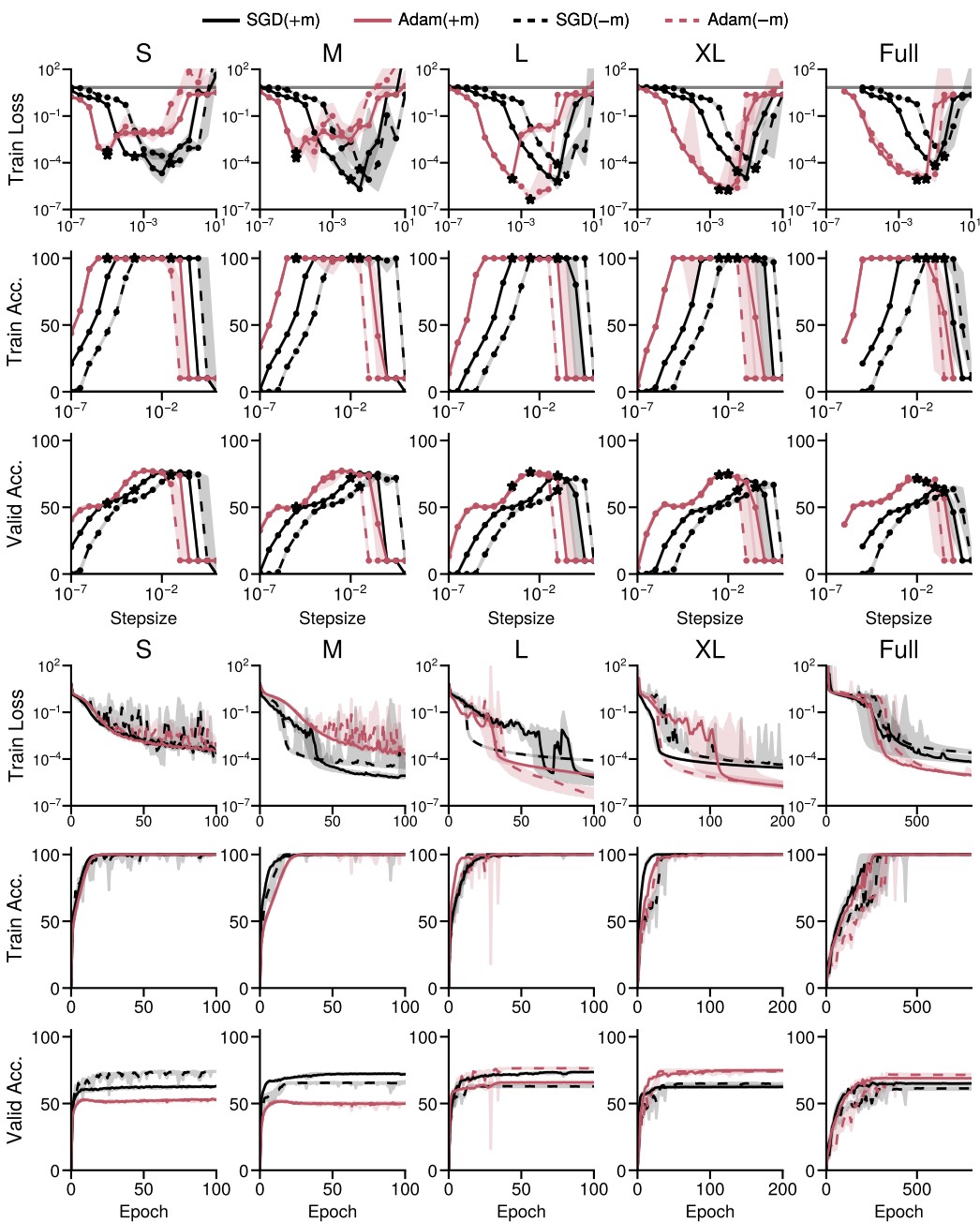

Figure 13: **Grid search and selected runs for each batch size ResNet18 on Cifar10**. **Top:** Loss and evaluation metrics at the end of training (see Table 1) for step-sizes evaluated. Grey line corresponds to value at initialization. **Bottom:** Performance of the selected step-size on training loss and evaluation metrics during training.

### C.1.3 SMALL TRANSFORMER ON PTB

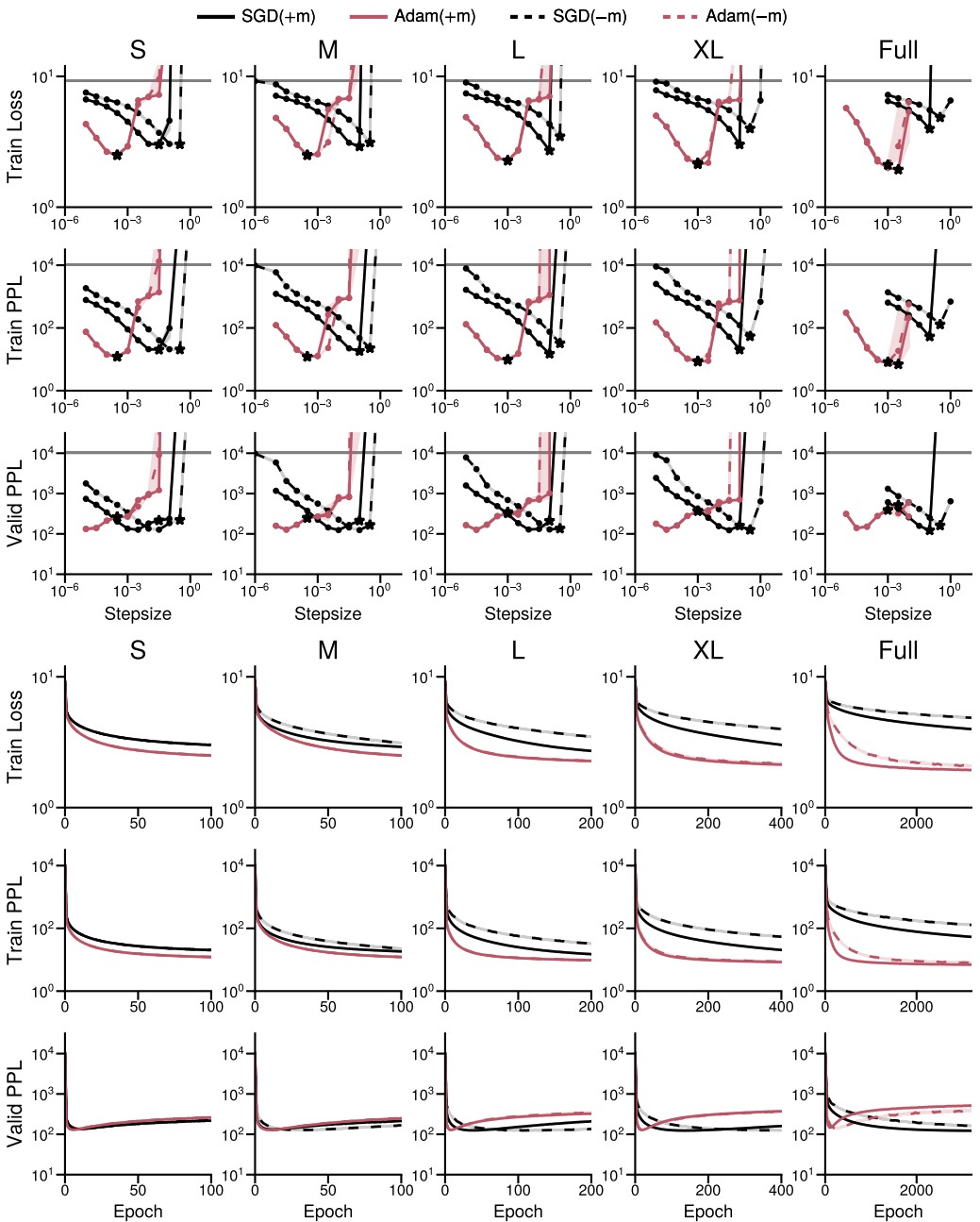

Figure 14: **Grid search and selected runs for each batch size Small Transformer on PTB. Top:** Loss and evaluation metrics at the end of training (see Table 1) for step-sizes evaluated. Grey line corresponds to value at initialization. **Bottom:** Performance of the selected step-size on training loss and evaluation metrics during training.

### C.1.4 TRANSFORMER-XL ON WIKITEXT-2

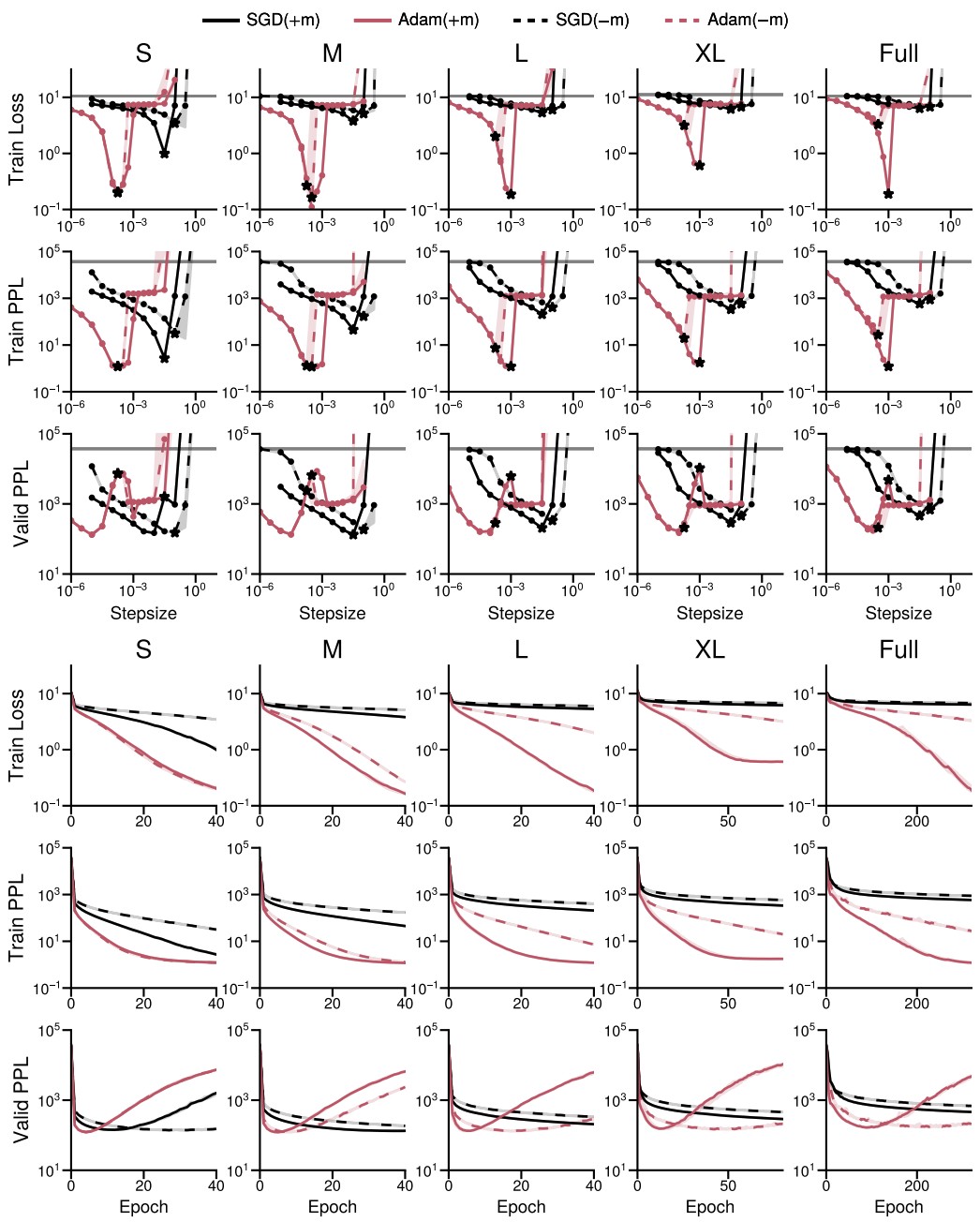

Figure 15: **Grid search and selected runs for each batch size Transformer-XL on WikiText-2**. **Top:** Loss and evaluation metrics at the end of training (see Table 1) for step-sizes evaluated. Grey line corresponds to value at initialization. **Bottom:** Performance of the selected step-size on training loss and evaluation metrics during training.

## C.1.5 DISTILLBERT FINETUNING ON SQUAD

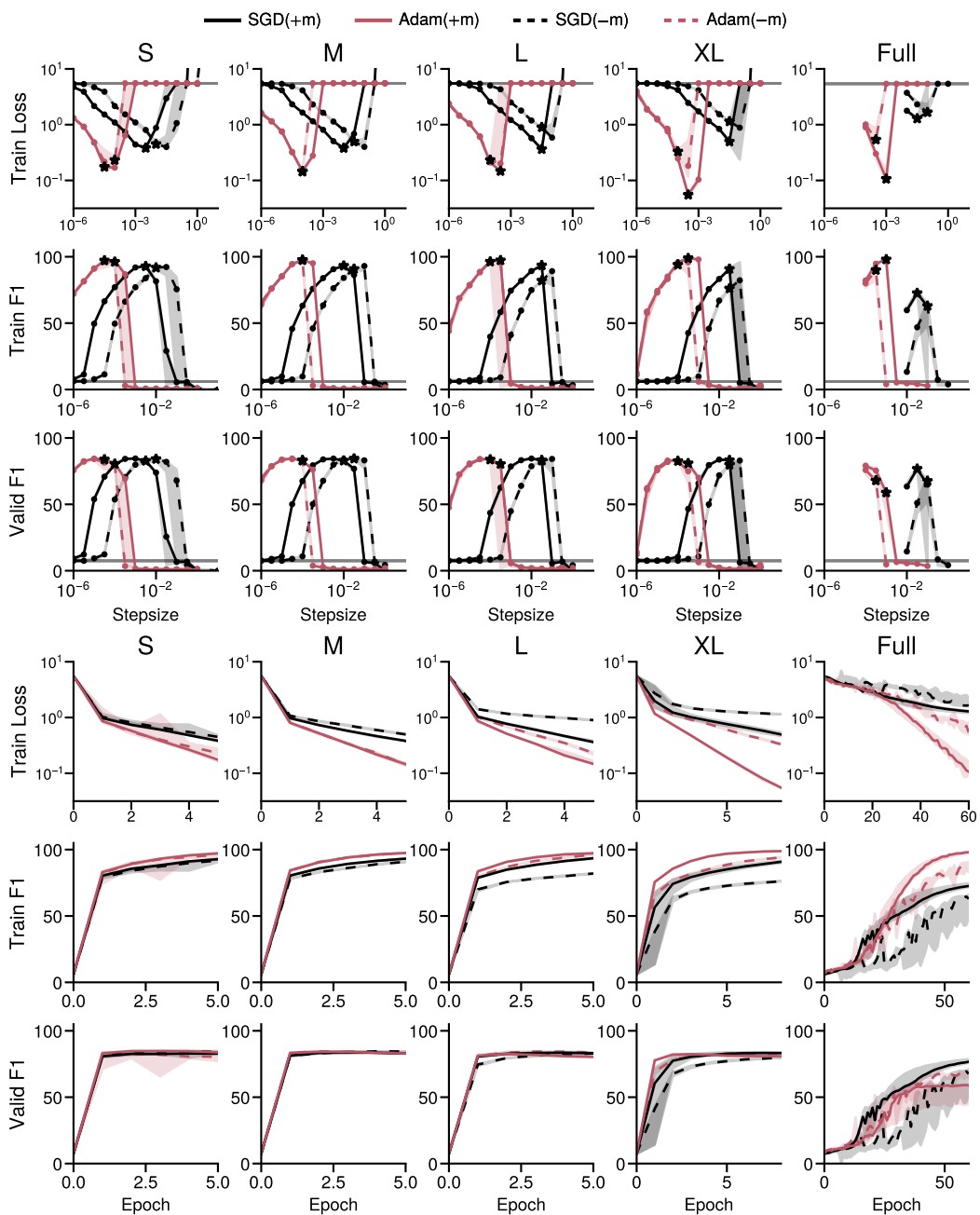

Figure 16: **Grid search and selected runs for each batch size DistillBERT finetuning on SQuAD**. **Top:** Loss and evaluation metrics at the end of training (see Table 1) for step-sizes evaluated. Grey line corresponds to value at initialization. **Bottom:** Performance of the selected step-size on training loss and evaluation metrics during training.

## C.2  SIGN DESCENT AND NORMALIZED GD

### C.2.1  MODIFIED LENET5 ON MNIST

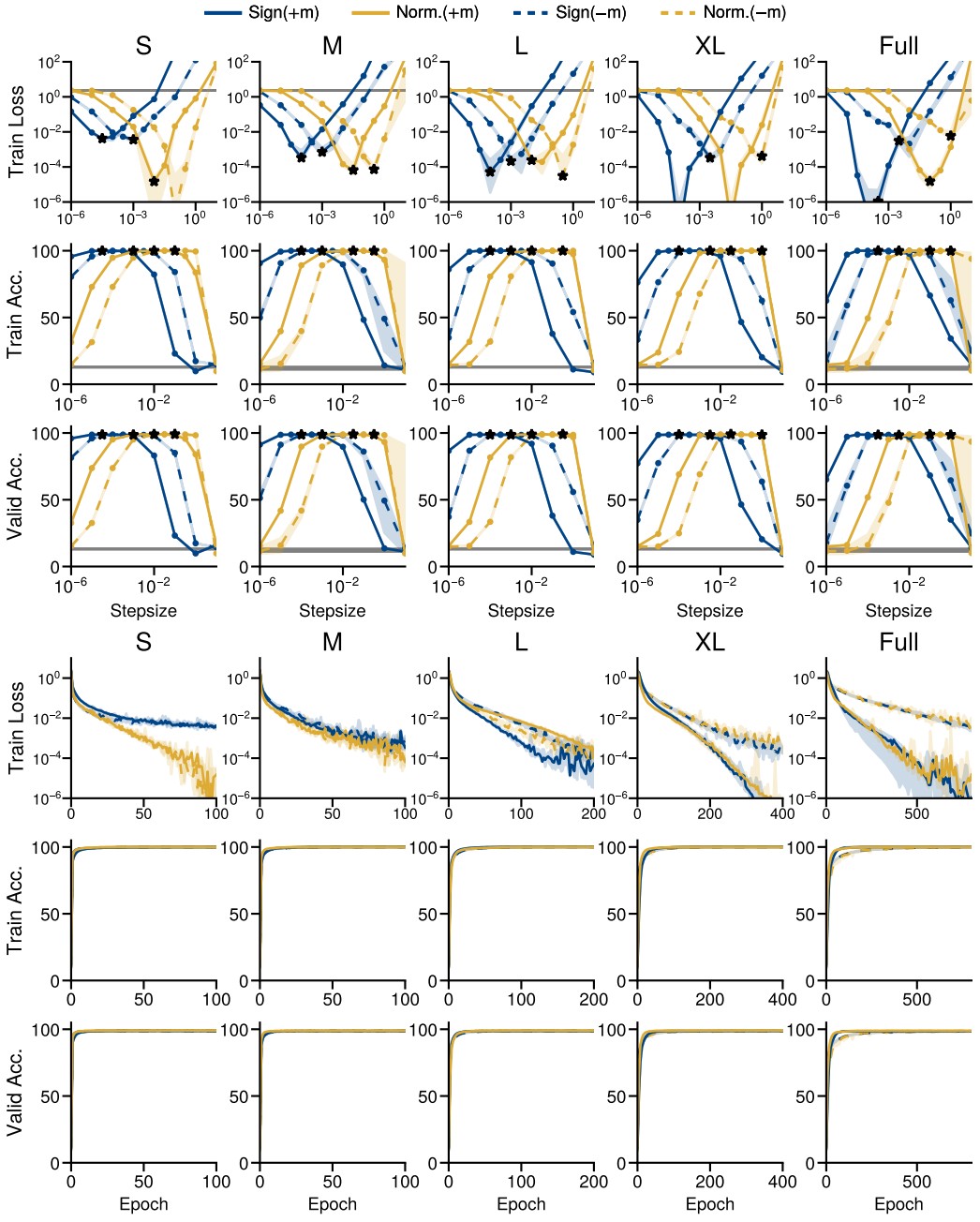

Figure 17: **Grid search and selected runs for each batch size modified LeNet5 on MNIST**. **Top:** Loss and evaluation metrics at the end of training (see Table 1) for step-sizes evaluated. Grey line corresponds to value at initialization. **Bottom:** Performance of the selected step-size on training loss and evaluation metrics during training.

## C.2.2 RESNET18 ON CIFAR10

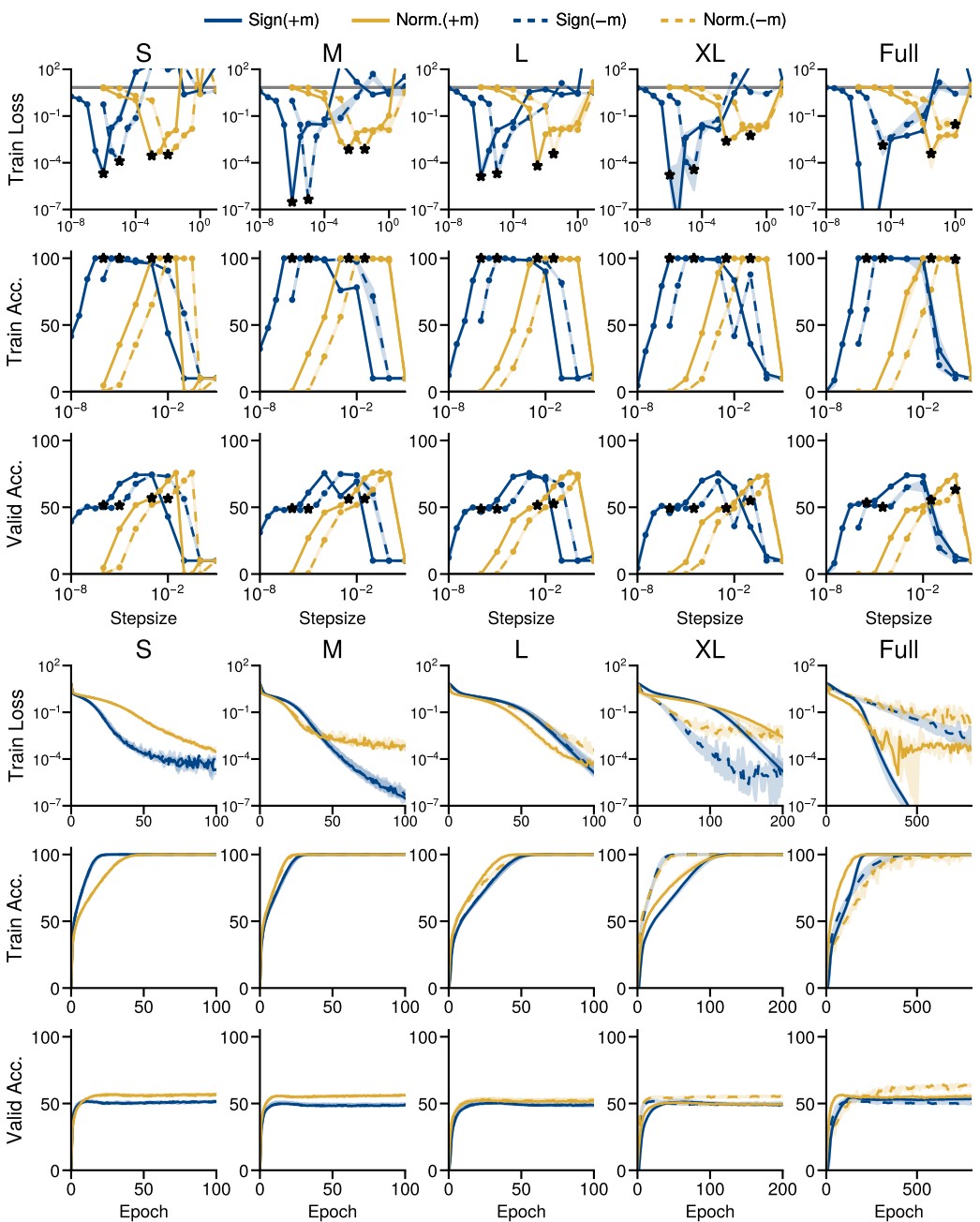

Figure 18: **Grid search and selected runs for each batch size ResNet18 on Cifar10**. **Top:** Loss and evaluation metrics at the end of training (see Table 1) for step-sizes evaluated. Grey line corresponds to value at initialization. **Bottom:** Performance of the selected step-size on training loss and evaluation metrics during training.

### C.2.3   SMALL TRANSFORMER ON PTB

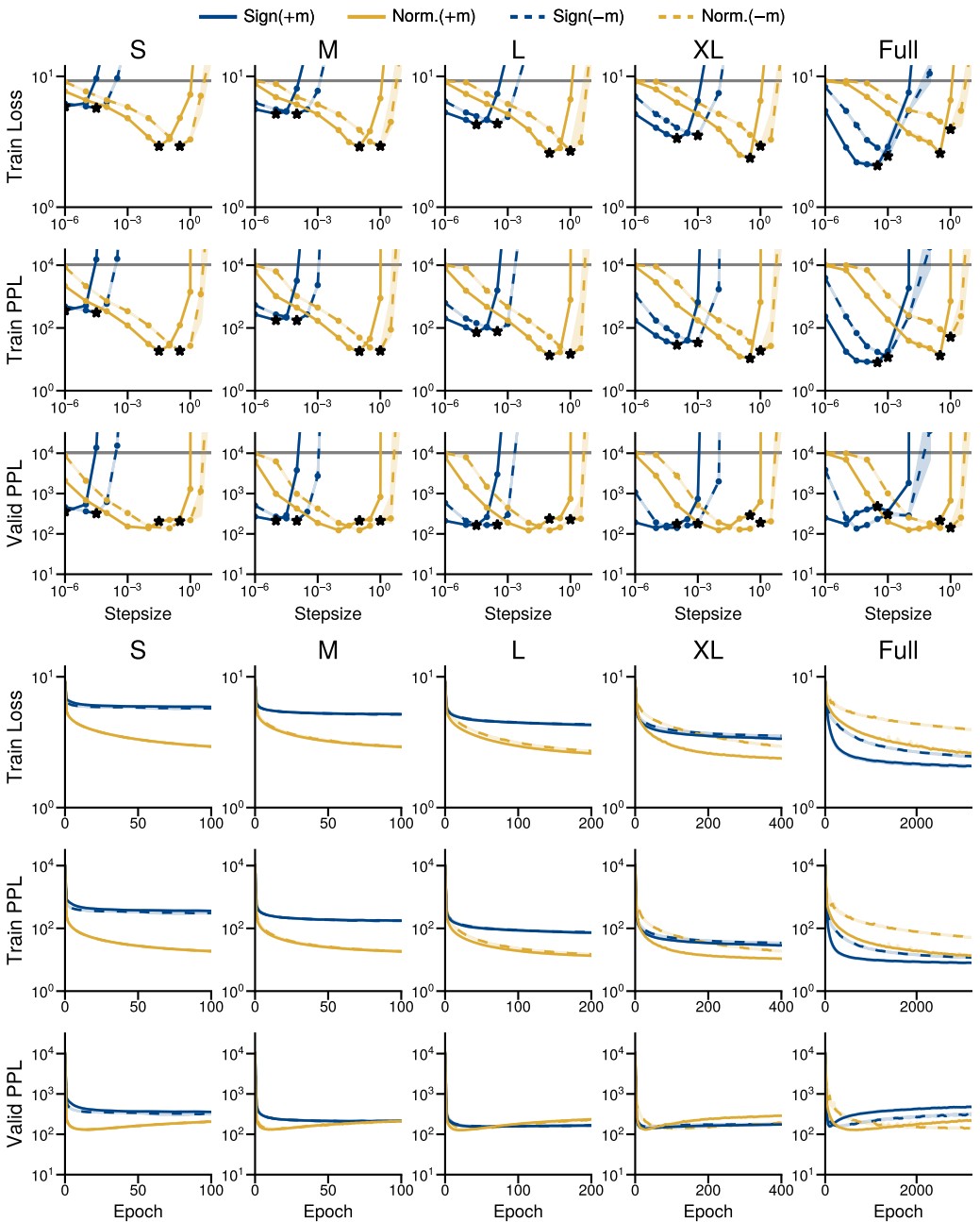

Figure 19: **Grid search and selected runs for each batch size Small Transformer on PTB. Top:** Loss and evaluation metrics at the end of training (see Table 1) for step-sizes evaluated. Grey line corresponds to value at initialization. **Bottom:** Performance of the selected step-size on training loss and evaluation metrics during training.

### C.2.4 TRANSFORMER-XL ON WIKITEXT-2

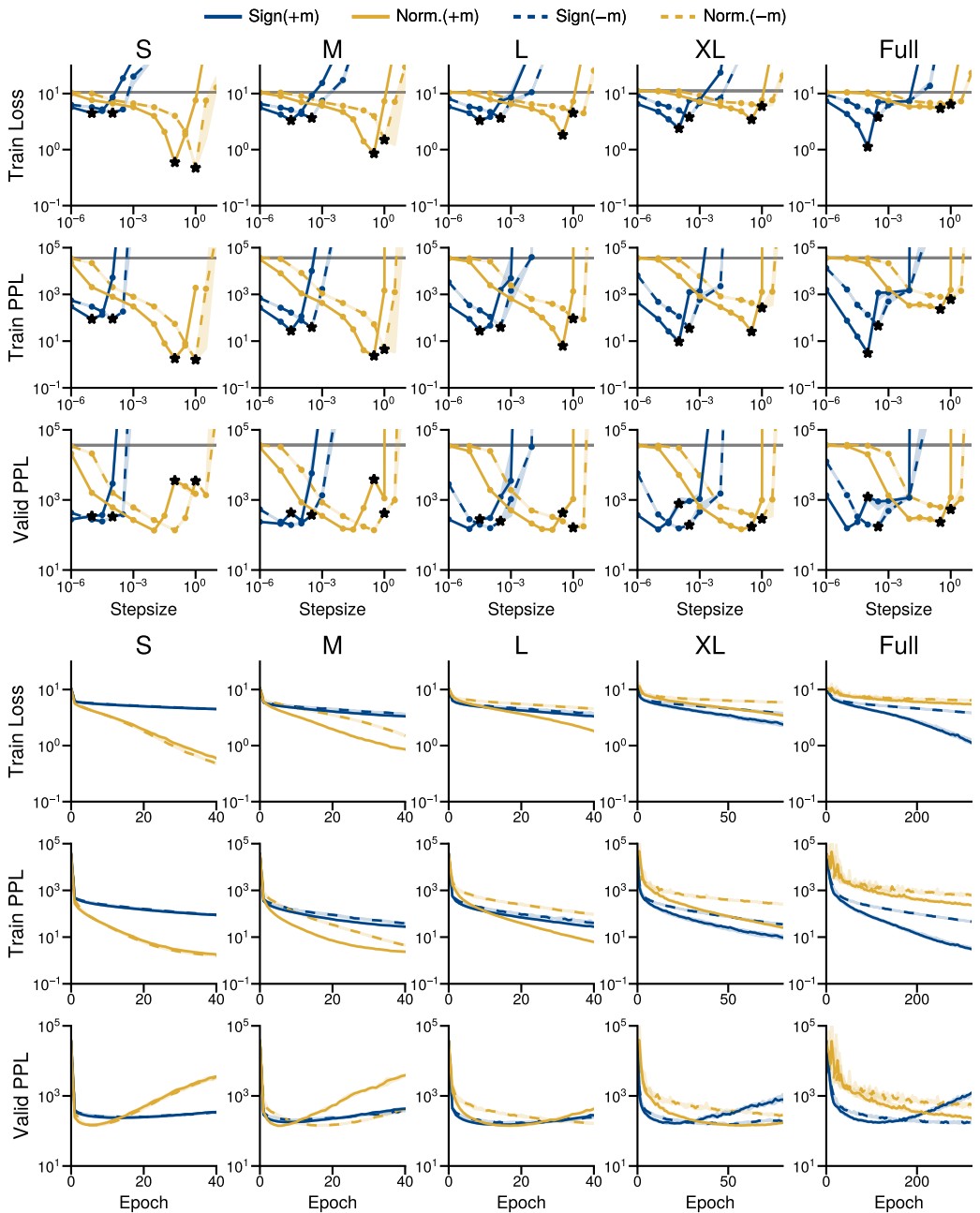

Figure 20: **Grid search and selected runs for each batch size Transformer-XL on WikiText-2**. **Top:** Loss and evaluation metrics at the end of training (see Table 1) for step-sizes evaluated. Grey line corresponds to value at initialization. **Bottom:** Performance of the selected step-size on training loss and evaluation metrics during training.

### C.2.5 DISTILLBERT FINETUNING ON SQUAD

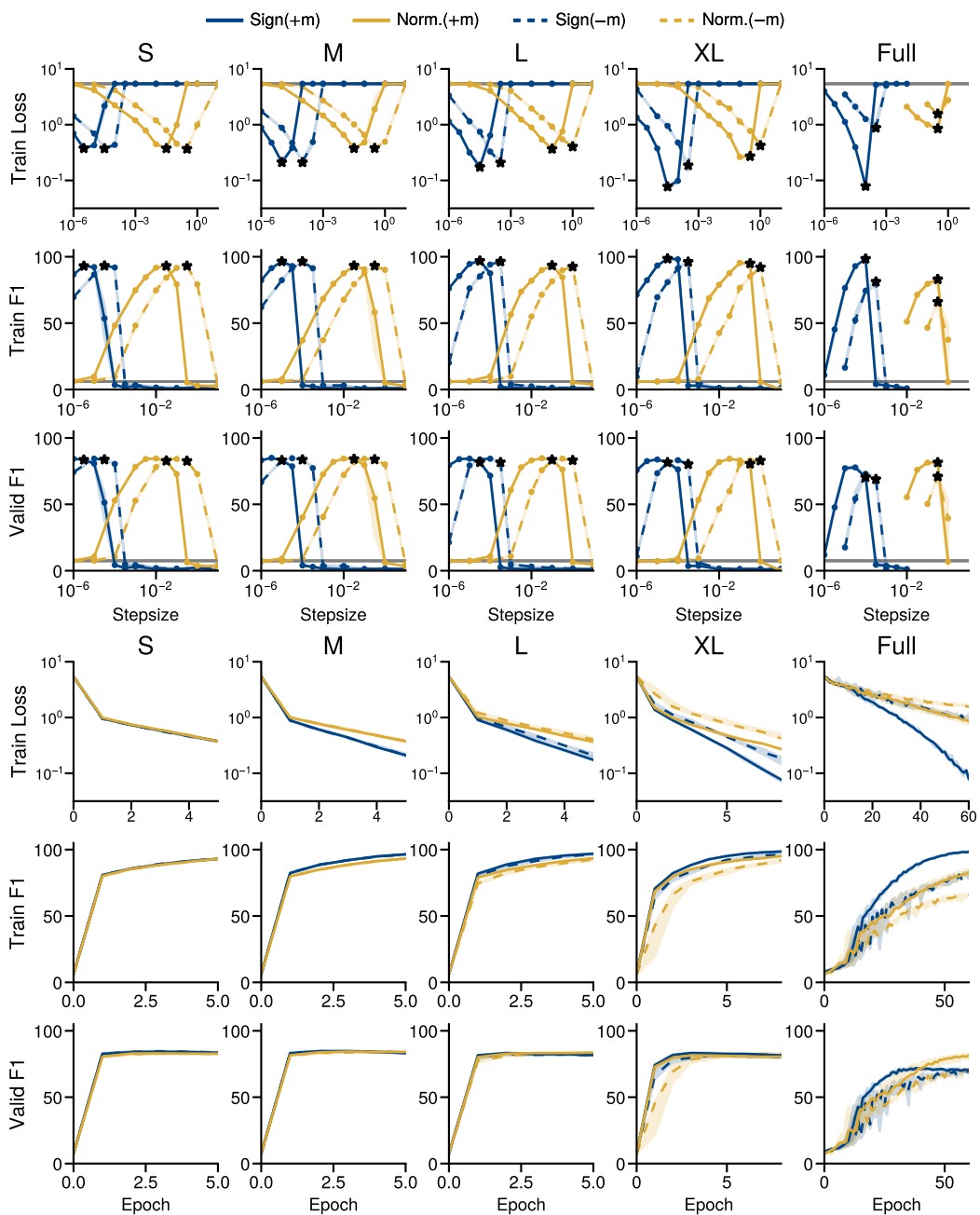

Figure 21: **Grid search and selected runs for each batch size DistillBERT finetuning on SQuAD**. **Top:** Loss and evaluation metrics at the end of training (see Table 1) for step-sizes evaluated. Grey line corresponds to value at initialization. **Bottom:** Performance of the selected step-size on training loss and evaluation metrics during training.

