# OpenReview forum: "Noise Is Not the Main Factor Behind the Gap Between Sgd and Adam on Transformers, But Sign Descent Might Be"
_ICLR.cc/2023/Conference — ICLR 2023 poster_

### Official Review · Reviewer_J7tc · 2022-10-22

**Confidence:** 4
**Correctness:** 4
**Technical Novelty And Significance:** 3
**Empirical Novelty And Significance:** 3
**Recommendation:** 6

**Clarity, Quality, Novelty And Reproducibility:**

**Clarity**

The writing of this paper is clear and I am able to the key information without much effort.

**Novelty**

To the best of my knowledge, this is the first work investigating the correctness of the adaptivity-to-noise hypothesis.

**Strength And Weaknesses:**

**Strength**

1. The question that the paper aims to study may be of interest to the optimization community. The adaptivity-to-noise hypothesis is impactful and its correctness is worth studying. Ruling out this hypothesis also indicates that new hypotheses need to be considered.

2. The experiments are extensive and solid, thus convincing.

**Weakness**

1. While disproving the adaptivity-to-noise hypothesis, this paper does not provide a new hypothesis/conjecture. It is unclear why signGD works well in the large batch regime and why signGD works badly in the small batch regime.

**Questions**

In (Zhang et al. 2019), it is hypothesized that adaptive methods perform better than SGD since the smoothness is not uniformly bounded but can be controlled by the gradient norm. It also proves that normalized GD can adapt to such a landscape and converge arbitrarily faster than SGD. If this was the key mechanism of the acceleration effect of adaptive methods, then clipped SGD should perform as well as Adam in the full-batch case. However, this paper shows that normalized GD performs much worse than Adam in the full-batch case. Does this mean this hypothesis is also wrong?

**References**

Zhang et al. Why gradient clipping accelerates training: A theoretical justification for adaptivity, 2019


**Summary Of The Paper:**

**Overview of this paper**

This paper studies why adaptive methods perform better than SGD in terms of convergence. Specifically, this paper aims to explore the hypothesis proposed by (Zhang et al., 2019): Adaptive methods including Adam converge faster than SGD due to the more robust gradient estimate of adaptive methods (abbreviated as the adaptivity-to-noise hypothesis latter). Through extensive experiments, it is observed that the convergence gap between Adam and SGD even gets larger when the batch size increases (and thus the noise decreases), and thus the adaptivity to the noise of adaptive optimizers can not fully explain the gap.

**Summary of the experiment observations**

(1). Over language tasks, the training loss gap between SGD and Adam (both best tuned) gets larger when the batch size increases.

(2). Adding momentum improves the performance of optimizers, mainly in the case when the batch size is large.

(3). The performance of SignGD in the small-batch regime, but improves significantly with respect to batch. Over some tasks (e.g., PTB), the performance of full-batch SignGD is comparable to that of full-batch Adam.

(4). Normalized gradient descent also scales better with batch size than plain gradient descent, but less so than sign descent. Full-batch normalized gradient descent performs worse than full-batch SignGD.



**References**

Zhang et al.,  Why are adaptive methods good for attention models?, 2019

**Summary Of The Review:**

This work is the first to investigate the correctness of the adaptivity-to-noise hypothesis, which is an impactful hypothesis for why Adam converges faster than SGD. While this paper does not provide a new hypothesis, it disproves the adaptivity-to-noise hypothesis through extensive and sound experiments. I believe this result is of interest to the optimization-in-deep-learning community, and lean on acceptance of this work.

---

> ### Author Response · Authors · 2022-11-14
> **Initial reply to Reviewer J7tc**
>
> Thank you for your review. We discuss the issues brought up below.
>
> > While disproving the adaptivity-to-noise hypothesis, this paper does not provide a new hypothesis/conjecture. It is unclear why signGD works well in the large batch regime and why signGD works badly in the small batch regime.
>
> The poor behavior of SignDescent in the stochastic setting could be attributed to noise, but the good behavior in the deterministic setting is indeed not well understood. While we do not provide a theoretical model of this behavior, our contribution is to highlight this unexplained empirical behavior in the full batch setting, and provide evidence in favor of a deterministic interpretation of the benefit of Adam over SGD.
>
> > In (Zhang et al. 2019), it is hypothesized that adaptive methods perform better than SGD since the smoothness is not uniformly bounded but can be controlled by the gradient norm. It also proves that normalized GD can adapt to such a landscape and converge arbitrarily faster than SGD. If this was the key mechanism of the acceleration effect of adaptive methods, then clipped SGD should perform as well as Adam in the full-batch case. However, this paper shows that normalized GD performs much worse than Adam in the full-batch case. Does this mean this hypothesis is also wrong?
>
> This is a nice insight and we agree that the non-uniform bound alone is insufficient. We discuss this assumption and alternatives in section 6.2 and highlight that even though Zhang et al. propose the relaxed smoothness assumption, they assume it holds element-wise to build an element-wise clipping method that is closer to Sign Descent than Normalized GD. We have changed this paragraph to more explicitly state that relaxed smoothness/clipping is insufficient, and that element-wise clipping/sign descent might be a better alternative.

---

> > ### Comment · Reviewer_J7tc · 2022-11-22
> > **Thanks for your response**
> >
> > I want to thank the authors for the response and would like to keep my score.
> >
> > Additional discussion on sign descent ( this will not affect my score): element-wise smoothness assumption is strictly stronger than the non-element-wise one. Therefore, it seems that clipped sgd also converges under the element-wise smoothness assumption. So how to distinguish sign sgd and clipped sgd in this case?

---

> > > ### Author Response · Authors · 2022-12-09
> > > **Additional response**
> > >
> > >
> > > Thanks for your response!
> > >
> > > > element-wise smoothness assumption is strictly stronger than the non-element-wise one. Therefore, it seems that clipped sgd also converges under the element-wise smoothness assumption. So how to distinguish sign sgd and clipped sgd in this case?
> > >
> > > Indeed, current works show that clipped-GD is a good fit for functions satisfying relaxed smoothness and element-wise clipping for the element-wise form of the assumption, which is similar to sign descent. The two conditions are similar, but the element-wise one is much more complex, making the distinction difficult to investigate, and we believe it is outside of the scope of our submission.
> > >
> > > Crawshaw et al. (2022) argue that global relaxed smoothness does not capture all the complexity of the loss surface, because Figure 3 (in their work) shows that the smoothness along coordinates in different layers have different scaling behavior with the gradient of that coordinate. In the last layer of the network, the measured smoothness scales with the norm of the gradient, while the scaling is flatter for earlier layers.
> > >
> > > One path to better understand the differences between assumptions might be to find a less extreme version of relaxed smoothness that still explains this behavior. For example, if the key difference is between the last layer and the rest of the network, a two-block split, rather than element-wise, would be more verifiable.
> > >
> > > (We notice a misattribution error that will be corrected; In the last paragraph of Section 5, we attribute the coordinate-wise version of clipping to Zhang et al. (2020a) and Crawshaw et al. (2022) – this is incorrect and it should be attributed solely to Crawshaw et al. (2022))

---

### Official Review · Reviewer_AS9c · 2022-10-27

**Confidence:** 4
**Correctness:** 3
**Technical Novelty And Significance:** 2
**Empirical Novelty And Significance:** 3
**Recommendation:** 6

**Clarity, Quality, Novelty And Reproducibility:**

Clarity:
The overall layout of the paper is very clear, and I appreciated that the paper has a clear narrative. However I would encourage the authors to improve the presentation of the figures. I found it difficult to quickly determine which curve corresponded to which batch size/algorithm. The text is also quite long-winded in places, and I'd encourage the authors to make the writing more succinct.

The performance of Adam and Sign Descent is compared in Figures 3/4 without plotting both curves on the same plot.

Quality:
I think the paper does a good job of conveying a simple core message with well chosen experiments. It was a shame that the authors did not study sign descent in more detail.

Novelty:
The core ideas have mostly appeared in prior work, but since the paper is primarily empirical I think that is fine.

Reproducibility:
I think the paper could be reproduced

Other comments:

1) Why does the "Bad starting assumptions can lead us astray" paragraph appear in the intro? It didn't feel very relevant to the paper to me.
2) On heavy tailed noise: note that heavy tailed noise likely arises when the Hessian is poorly conditioned, and this may also explain why sign descent outperforms gradient descent. Ie heavy tailed gradients and sign descent outperforming gradient descent may well have a common cause.
3) Zhang et al. (https://arxiv.org/abs/1907.04164) previously showed that Adam scales to larger batch sizes than SGD, while Zhang et al., Smith et al. (https://arxiv.org/pdf/2006.15081.pdf) and Shallue et al. (https://arxiv.org/abs/1811.03600) all showed that Momentum scales to larger batch sizes.
4) Related to point 2, Zhang et al. and Smith et al. both argued that batch size scaling is connected to the conditioning of the Hessian. It would be nice to see some discussion of this/other explanations for the success of sign descent, especially with regards to the differences between vision and language tasks.


**Strength And Weaknesses:**

Strengths:
1) The paper tells a clear story, and makes a convincing case that the key differences between Adam and SGD are best studied in the large batch/low noise regime.
2) The observation that sign descent outperforms gradient descent in the full batch limit is quite interesting, and provides a plausible intuition for the success of Adam.

Weaknesses:
1) The sign descent experiments are not very convincing. There is still quite a significant gap between sign descent and Adam in the large batch regime on some datasets (eg Figure 5 wikitext/Squad).
2) The authors note that the gap between SGD and Adam primarily arises on language/transformer tasks, however this observation is not explored/discussed.

**Summary Of The Paper:**

The authors show that across a range of tasks, the performance of Adam improves more as the batch size rises than the performance of SGD. In the full batch limit, Adam continues to significantly outperform SGD on language tasks. Based on this observation, they argue that the benefits of Adam are best studied in the full-batch/deterministic limit, and that key benefit of Adam is therefore not better robustness in the presence of heavy tailed gradient noise.

Additionally, they observe that sign descent also improves considerably as the batch size rises, and that while it performs poorly for small batch sizes it achieves comparable performance to Adam on some tasks in the full batch limit. They therefore propose that studying sign descent in the full batch limit may be a useful toy model for understanding the benefits of Adam.

**Summary Of The Review:**

I think this paper makes a valuable contribution to the field. However I think it could be significantly improved by improving/extending the experiments and discussion of sign descent. I therefore will score it weak accept for now.

---

> ### Author Response · Authors · 2022-11-14
> **Initial reply to Reviewer AS9c**
>
> Thank you for your review. We discuss the issues brought up below.
>
> > The sign descent experiments are not very convincing. There is still [a gap] on some datasets (eg Figure 5 wikitext/Squad).
>
> We agree this is a point we should be clearer on, and a similar point was raised by reviewer nxBD.
>
> We agree that Contribution 3 is an overstatement as written (“The performance of Adam is very similar to sign descent in full batch”). We have changed it to “Sign descent can close most of the gap between SGD and Adam in full batch”. This description summarizes the observations that Sign Descent closes most of the gap between SGD and Adam on some problems, shown in Figure 5 (when all methods use momentum) and all problems when run without momentum (Fig. 8). We do not expect Sign Descent to perform as well as Adam; our experiment is meant to determine which algorithmic component of Adam contributes to the positive scaling with batch size. We have reworded the final paragraph of Section 5 to be more explicit.
>
> (That the performance of Adam does not degrade so drastically in the small batch setting is consistent with the interpretation of Adam or RMSProp as a form of smoothed sign descent for the stochastic setting, despite formal justification of this effect in those works, or a justification for why sign descent is a good idea in the deterministic setting.)
>
> > The authors note that the gap between SGD and Adam primarily arises on language/transformer tasks, however this observation is not explored/discussed.
>
> We do not yet have a good understanding of why this behavior is more visible on transformers than vision models (this observation was made by Zhang et al. (2019), as described in the introduction). Our goal is to narrow down the source of the gap between SGD and Adam to help answer this question.
>
> > [...] heavy tailed noise likely arises when the Hessian is poorly conditioned, and this may also explain why sign descent outperforms gradient descent. [They] may well have a common cause.
>
> > Related to point 2, Zhang et al. and Smith et al. both argued that batch size scaling is connected to the conditioning of the Hessian. It would be nice to see some discussion of this/other explanations for the success of sign descent, especially with regards to the differences between vision and language tasks.
>
> We agree that there might be a correlation between poor conditioning and heavy-tailed gradients, and they might have a common cause. However current models of batch size scaling that assume the function is quadratic do not explain why sign descent outperforms gradient descent on poorly conditioned problems. We discuss these issues in the conclusion section, along with assumptions that relax smoothness to depend on gradient norm as possible directions to explain why sign descent helps with conditioning.
>
> > Zhang et al. previously showed that Adam scales to larger batch sizes than SGD, while Zhang et al., Smith et al. and Shallue et al. all showed that Momentum scales to larger batch sizes.
>
> We agree, and do discuss the contributions of Zhang et al. and Shallue et al. Please let us know if you feel we have missed discussing an important aspect of their work.
>
> The work of Smith et al. is primarily concerned with the effect of batch size on generalization, and is less relevant to our work than those of Zhang et al. and Shallue et al, as we focus on the modeling of the optimization behavior on training error.
>
> > Why does the "Bad starting assumptions can lead us astray" paragraph appear in the intro? It didn't feel very relevant to the paper to me.
>
> Our work attempts to give evidence on what is wrong with the “models” used in optimization (e.g. smooth functions and gradient estimators with bounded variance), to improve our starting assumptions. This paragraph motivates this line of research by highlighting what can go wrong when assumptions are not satisfied, but conclusions from theory are incorporated into common knowledge anyway. We believe this is relevant to our objective, and welcome suggestions to make this message clearer to the reader.
>
> > However I would encourage the authors to improve the presentation of the figures. I found it difficult to quickly determine which curve corresponded to which batch size/algorithm.
>
> We have added a simpler visualization of the final performance vs. the batch size in Figure 20 and will try to improve the annotations.
>
> > It was a shame that the authors did not study sign descent in more detail.
>
> We agree that sign descent is a very interesting avenue for understanding the performance of Adam on stochastic and/or poorly conditioned problems. At the same time, our study is primarily focused on the performance gap between Adam and SGD, so that an in-depth study of sign descent is outside the paper's scope. We are running another round of experiments and would be happy to expand on sign descent if there are particular properties you feel we should test more thoroughly.

---

> > ### Comment · Reviewer_AS9c · 2022-11-23
> > **thanks for your response**
> >
> > Thanks for your response. A few minor comments below:
> >
> > 1) Thank you for agreeing to soften the claims regarding sign-descent.
> >
> > 2) Is it a known result that sign-SGD doesn't outperform SGD on the noisy quadratic model (NQM) for large batch sizes? Given that Zhang et al. show that Adam does outperform SGD on the NQM for large batch sizes, I would expect sign-SGD to do the same?
> >
> > 3) See figure 1 in Smith et al.: this shows that Momentum only outperforms SGD (on both train and test set) when the batch size is large. I think these three works (Shallue, Zhang, Smith) should be cited alongside claim 3.c in the text ("The importance of momentum is increased in full batch"). I suspect there are also other papers that have shown this result I'm not aware of.
> >
> > 4) I personally don't think the "Bad starting assumptions" paragraph adds to the text currently because no explicit link is made to the core claims of the paper. I think this paragraph would work better if the text clarified why the authors think this paper is an example of this error.
> >
> > 5) For clarity, I agree that the paper doesn't require a theoretical study of sign descent and this might be out of scope, but alternatively a more detailed empirical evaluation of how similar Adam and sign descent are in the large/full batch setting would strengthen the paper.

---

> > > ### Author Response · Authors · 2022-12-09
> > > **Additional response**
> > >
> > >
> > > Thanks for your response!
> > >
> > > > 2.  Is it a known result that sign-SGD doesn't outperform SGD on the noisy quadratic model (NQM) for large batch sizes? Given that [...] Adam does outperform SGD on the NQM for large batch sizes, I would expect sign-SGD to do the same?
> > >
> > > Zhang et al. [arXiv/1907.04164](https://arxiv.org/pdf/1907.04164.pdf) do not provide evidence that Adam outperforms SGD on the NQM. They show that preconditioning by the inverse Hessian outperforms gradient descent and scales better with batch size (Figure 3) and argue that this is an explanation for the benefit of Adam and K-FAC, as those methods also use preconditioning (item 3 in their introduction). Our argument is that this is an insufficient model, as the interpretation of Adam as Sign Descent is inconsistent with “preconditioning with the Hessian” on quadratic models. We will rephrase the paragraph to make this point clearer.
> > >
> > > On SignDescent vs. GD on quadratics, the two algorithms behave very differently, which makes the comparison difficult. It is possible to find settings where SignDescent is competitive if the step-size is tuned appropriately, and settings where GD arbitrarily outperforms SignDescent. Consider the toy example `f(x) = .5‖x - y‖²`. GD with a step-size of 1 solves the problem in one step, regardless of initialization. The best step-size for Sign Descent heavily depends on initialization because the step does not scale with the norm of the gradient. Due to this issue, there is little theory on (non-modified) Sign Descent (it is not guaranteed to converge with a constant step-size). The magnitude of the update does not go to 0 at the minimum and the algorithm can bounce around. Theory typically analyzes a rescaled variant of sign descent, with the update `‖∇f(x)‖₁ sign(∇f(x))`. Rescaled Sign Descent can outperform GD on some quadratics, depending on the distribution of the eigenvalues and the rotation of the Hessian (see e.g. Fig. 2 in studied by Balles et al., 2020 - [(arXiv/2002.08056)](https://arxiv.org/pdf/2002.08056.pdf)). The NQM of Zhang et al. consider diagonal Hessians, which is the most favorable case for (rescaled) sign descent.
> > >
> > > > 3.  See figure 1 in Smith et al.: [...] I think these three works (Shallue, Zhang, Smith) should be cited alongside claim 3.c in the text [...]
> > >
> > > We agree; we will move the citations to Section 3 after 3.c (currently in Section 4 after 4.b) and add a citation to Smith et al.
> > >
> > > > 4.  I personally don't think the "Bad starting assumptions" paragraph adds to the text currently because no explicit link is made to the core claims of the paper. I think this paragraph would work better if the text clarified why the authors think this paper is an example of this error.
> > >
> > > Thanks for clarifying, we agree that connection should be made explicit. We will reword it.
> > >
> > > > 5.  For clarity, I agree that the paper doesn't require a theoretical study of sign descent and this might be out of scope, but alternatively a more detailed empirical evaluation of how similar Adam and sign descent are in the large/full batch setting would strengthen the paper.
> > >
> > > We agree that a better understanding of this similarity would be beneficial, but were already surprised at how close the performance of Sign Descent was to Adam in full batch. We welcome suggestions on specific experiments to try or hypotheses to test to include in a revision.

---

### Official Review · Reviewer_nxBD · 2022-11-01

**Confidence:** 4
**Correctness:** 2
**Technical Novelty And Significance:** 3
**Empirical Novelty And Significance:** 3
**Recommendation:** 5

**Clarity, Quality, Novelty And Reproducibility:**

**Clarity** Some part of the paper are not clear (e.g., plots)

**Quality** the paper has some problems such as overclaiming

**Novelty** the paper is novel

**Reproducibility** Seems good.


**Strength And Weaknesses:**

### Strength
- The paper’s motivation to establish the correct assumptions for theoretical work on optimization is refreshing and extremely important.
- The experimental procedure is carefully designed

### Weakness
In my opinion, the major weakness of the paper is overclaiming, especially with regard to the claim about refuting “long-tailed gradient error is why Adam does better than SGD”. I will comment on each of the claims made individually.

**Claim 1**: comparing figure 1 and figure 2, it does seem Adam benefits from the full batch more than SGD. This claim seems well supported.

**Claim 2**: This claim only seems to hold for a subset of the experiments and does not say anything about the long-tailed nature of SGD. Further, even in Figure 3, I am not sure how I should reach the conclusion “SGD does not take advantage of the reduction in noise while Adam does”. On Squad, it seems that SGD is able to take advantage of the increased batchsize. Similarly, on PTB, SGD seems to do better when the batch size increases. The argument seems strenuous at best. Finally, in Figure B.2, it seems that Mnist and Cifar10 exhibit very different behavior even though both are supposed to not have long-tailed gradient noise according to figure 1. This casts further doubt on the claim about long-tailed gradient error.

**Claim 3**: This claim does not seem supported by the figures. In figure 4, I am not sure how to get “signed GD scales better than normalized GD” from the plots. In particular, it seems to me that signed GD since the larger batch sizes have higher training loss. If I am misinterpreting the plots, please correct me and add appropriate instructions about how to read the plots.

Overall, I find the plots to be difficult to read even with the help of the text. The addition of two sets of experiments for w/ and w/o momentum makes the matter worse. The momentum and no momentum plots (e.g., figure 3) should ideally be separate from each other. The dashed lines look really bad presumably because of the fact that the lines are sometimes not smooth. I would rather see a plot of the final performance against the number of each batch size rather than seeing their progress which doesn’t add anything. I think it would be more helpful to quantify how much the batchsize is correlated with training loss rather than commenting on it qualitatively.

More importantly, I believe that the paper did not fully disprove the hypothesis about heavy-tailed gradient error. In particular, I believe that what the paper showed is that “gradient noise does not explain the gap between Adam and SGD '' by showing that there is still a gap between the two even using fullbatch GD. This is somewhat a subtle point, because the paper didn’t convincingly address the issue of heavy-tailedness noise but instead removed it altogether. There exist alternative hypotheses. “Full-batch improving adam” and “heavy tail worsening sgd” can coexist. For example, noise helps gradient descent in general but SGD performs worse when the noise is heavy-tailed, or gradient noise worsens Adam, but when there is noise, Adam performs better if the noise is heavy tail. If I am wrong about this, I would be more than happy to learn how the current set of experiments do not support these possibilities.

Given how much the paper focuses on the heavy-tailed gradient noise, I believe the experimental results do not fully support the claim. The findings are still interesting and it supports the claim that the difference between Adam and SGD is deterministic, but I don’t think they fully preclude the heavy-tailed hypothesis. I would like to emphasize that I do not believe in the hypothesis myself but I believe that it is important to be careful and rigorous in scientific claims. As such, I would suggest the authors rephrase the introduction and title accordingly to reflect subtleties.



**Summary Of The Paper:**

This work studies whether the heavy-tailed nature of gradient is the true cause of Adam outperforming SGD, as suggested by prior work (J. Zhang et. al., 20). The paper conducts an extensive set of experiments involving decreasing noise level of SGD/Adam via increasing the batchsize from small batch all the way to fullbatch GD. The paper makes three main observations: 1. Noise does not explain the gap between SGD and Adam since Adam still has lower training loss with full batch gradient, 2. SGD’s performance does not improve as the batchsize increases, which is a more granular version of claim 1, and 3. Adam’s success might be attributed to its behavioral similarities with signed gradient descent.

**Summary Of The Review:**

The paper makes interesting findings but makes some overclaims. I would increase my score to 6 or 8 if the authors can correct these overclaims and fix other issues that I mentioned above.

---

> ### Author Response · Authors · 2022-11-14
> **Initial reply to Reviewer nxBD**
>
> Thank you for your detailed review. We also think that rigorous hypothesis testing is critical to scientific discourse, which is why we attempt to include clear problem statements and careful reasoning throughout the paper. We appreciate the subtle difference between claiming that heavy-tailed noise has no effects on SGD and our intended main conclusion that heavy-tailed noise is not solely responsible for the gap between SGD and Adam. It will improve the paper to be more explicit about this difference. Many thanks for highlighting this issue.
>
> In what follows, we address specific concerns.
>
> > Claim 2: This claim only seems to hold for a subset of the experiments and does not say anything about the long-tailed nature of SGD. Further, even in Figure 3, I am not sure how I should reach the conclusion “SGD does not take advantage of the reduction in noise while Adam does”.
>
> Our intent is to not say that SGD cannot take advantage of a reduction in noise (it provably does under the right assumptions), but that Adam better takes advantage of the reduction in noise relative to SGD in these empirical settings. We have corrected Contribution 2 to “SGD benefits less from a reduction in noise than Adam”.
>
> > Claim 3: This claim does not seem supported by the figures. In figure 4, I am not sure how to get “signed GD scales better than normalized GD” from the plots.
>
> By “scale better” we meant to highlight the relative improvement with batch size was strongest for Sign Descent, possibly because it performs badly in absolute terms with small batches. We have edited the caption to clarify.
>
> Nevertheless, we agree that Contribution 3 is an overstatement as written (“The performance of Adam is very similar to sign descent in full batch”). We have changed it to “Sign descent can close most of the gap between SGD and Adam in full batch”. This description summarizes the observations that Sign Descent closes most of the gap between SGD and Adam on some problems, shown in Figure 5 (when all methods use momentum) and all problems when run without momentum (shown in Figure 8). We do not expect Sign Descent to perform as well as Adam; our experiment is meant to determine which algorithmic component of Adam contributes to the positive scaling with batch size. We have reworded the final paragraph of Section 5 to be more explicit.
>
> (That the performance of Adam does not degrade so drastically in the small batch setting is consistent with the interpretation of Adam or RMSProp as a form of smoothed sign descent for the stochastic setting, despite the lack of formal justification of this effect in those works, or a justification for why “sign descent” is a good idea to start with in the deterministic setting.)
>
> > Overall, I find the plots to be difficult to read even with the help of the text… I would rather see a plot of the final performance against the number of each batch size rather than seeing their progress which doesn’t add anything.
>
> We have thought carefully about how to visualize the phenomena that we observed in the experiments, and believe that the chosen plots give a lot more information about what is happening empirically beyond what would be observable in the final performance. For example, the similarities/differences in the trajectories can be observed in these plots, as well as the confounding factor that the different batch sizes run for a different number of iterations (performance could go up or down depending only on the number of iterations rather than the batch size). Nevertheless, we realize that these plots impose a high cognitive load on the reader so are planning to add a simpler visualization of the results too. We have added a visualization of final performance vs. batch size in Figure 20 (as to not change the figure numbers during the discussion period).
>
> > More importantly, I believe that the paper did not fully disprove the hypothesis about heavy-tailed gradient error. In particular, I believe that what the paper showed is that “gradient noise does not explain the gap between Adam and SGD '' by showing that there is still a gap between the two even using fullbatch GD. This is somewhat a subtle point, because the paper didn’t convincingly address the issue of heavy-tailedness noise but instead removed it altogether. There exist alternative hypotheses. “Full-batch improving adam” and “heavy tail worsening sgd” can coexist.
>
> “Stochasticity does not explain the gap between SGD and Adam” is the message we meant to convey. It is true that these alternative hypotheses can be true and would be consistent with our experiments, but our experiments do conclusively show that heavy-tailed noise (as a special case of gradient noise) cannot be the only explanation for the success of Adam. We believe our work will direct others towards more-promising explanations.

---

> > ### Comment · Reviewer_nxBD · 2022-11-18
> > **Thanks for the response**
> >
> > Thank you for updating the paper to be more rigorous. Overall, I still find the argument about sign gradient descent somewhat iffy and the interpretation needs more nuance. It would be good to have a quantitative measurement of how much more sign gradient is able to take advantage of batchsize rather than "interpreting" the plots. Further, there are still several places that still say SGD does not take advantage of reduction to noise so please fix them.
> > Nonethelss, since the main claims of the paper are now correct, I have increased my rating to 5.

---

> > > ### Author Response · Authors · 2022-12-09
> > > **Additional response**
> > >
> > > Thanks for your response!
> > >
> > > > Further, there are still several places that still say SGD does not take advantage of reduction to noise so please fix them.
> > >
> > > We will take a full pass on the submission to correct those statements.
> > >
> > > > It would be good to have a quantitative measurement of how much more sign gradient is able to take advantage of batchsize rather than "interpreting" the plots.
> > >
> > > We agree a quantitative measure would help. We are working towards quantifying "improvement per increase in batch size" in a way that does not suffer from confounding due to the different number of iterations.

---

### Official Review · Reviewer_emm1 · 2022-11-03

**Confidence:** 3
**Clarity, Quality, Novelty And Reproducibility:** The paper is well-written.
**Correctness:** 4
**Technical Novelty And Significance:** 3
**Empirical Novelty And Significance:** 3
**Recommendation:** 6

**Strength And Weaknesses:**

This paper studies the important problem of why Adam performs better than SGD (especially on language tasks), and provides interesting results. Prior work argues that the performance gap might be due to heavy tail of the noise distribution; however, this paper shows that heavy-tailed-ness  may not be enough to explain the performance gap, as it still exists even with full batches. Moreover, as the batch size increases, the SGD performance almost always degrades, while Adam can sometimes do better. Finally, with full batches, it is shown that sign descent can sometimes match the Adam performance. These findings can help us understand Adam better and design more efficient algorithms.

On the other hand, I have the following suggestions:
1. I think more hyperparameter tweaking is needed (e.g., some simple learning rate schedule). The reason is to make sure the results still hold when we have satisfactory test accuracies. For example, in Figure 11, the best test accuracy obtained for ResNet18 on CIFAR10 is below 80%, but I believe it should be easy to get 90%.
2. In Figure 5, to support the claim that Adam is similar to sign descent, it is also nice to compare the test error for Adam and sign descent.
2. For sign descent, in addition to moving averages of signs, it might also be interesting to consider signs of moving averages.

**Summary Of The Paper:**

This paper studies the performance gap between Adam and SGD at different batch sizes. It is found that the gap still exists with large batches or even full batches, which suggests the distribution of random noise may not explain this gap. By contrast, it is shown that Adam with large batches is similar to sign descent.

**Summary Of The Review:**

This paper provides interesting results, that the performance gap between Adam and SGD still exists with full batches, and that Adam is similar to sign descent. On the other hand, I think more experiments can be tried to make the story more complete. Therefore currently I put this paper marginally above the acceptance threshold.

---

> ### Author Response · Authors · 2022-11-14
> **Initial reply to Reviewer emm1**
>
> Thank you for your review. We discuss the issues brought up below.
>
> > I think more hyperparameter tweaking is needed (e.g., some simple learning rate schedule). [For example,] the best test accuracy obtained for ResNet18 on CIFAR10 is below 80%, but I believe it should be easy to get 90%.
>
> We agree that, barring computational constraints, test accuracy closer to state of the art would be preferable. However, while adding learning rate schedules might be sufficient for CIFAR10/ResNet18, it is much less clear on transformers. There is no standard learning rate schedule that we can use across problems, and other modifications are needed to obtain good results such as clipping. This would add additional confounders to our investigation on the effect of noise on the relative training performance of SGD vs. Adam.
>
> > In Figure 5, to support the claim that Adam is similar to sign descent, it is also nice to compare the test error for Adam and sign descent.
>
> We understand readers might be curious about the behavior on validation data. We provide this information alongside the grid-search validation in the appendix. However, we select the step-size according to the best training performance, which does not guarantee good or comparable performance on validation (p.3, Section 2: Experimental design, Hyperparameter tuning). Despite these limitations, the behavior of Adam and SignDescent on full batch runs does show similar behavior on validation data (by cross-checking figures, e.g. Fig 11 with Fig 17).
>
> > For sign descent, in addition to moving averages of signs, it might also be interesting to consider signs of moving averages.
>
> We use the moving averages of signs to ensure that the update “smoothes” the sign operation, as in RMSProp and Adam. Taking the sign of the moving average could lead to large jumps from small changes in the moving average, eg. from $\epsilon$ to $-\epsilon$, and less stable results. We did not include this method due to computational constraints. We are trying to be strategic about what experiments to run with our available resources, could you expand on what this experiment would be testing?

---

### Author Response · Authors · 2022-11-14
**General reply to all reviewers**

**We thank all reviewers for the work put in their reviews.**

As a primarily empirical contribution evaluating the validity of theoretical assumptions in optimization, we acknowledge that our contribution is non-standard. We are glad that this contribution was recognized and appreciated by all reviewers.

We thank the reviewers for recommendations on how to improve the clarity of the claims and the presentation, both in writing and visualizations. We take those comments seriously and will incorporate this feedback. The changes already mentioned in the responses here are highlighted in red in the updated pdf.

If we have missed a point, it is by mistake – please send us a reply on OpenReview, we are happy to expand on our answers.

**Update to the experiments**

We had the opportunity to run additional experiments since the deadline to investigate possible confounders and fix issues we found during our own review. We detail here the issues we have found and the solutions

- **Running SQuAD in full batch - No changes**

  We had an issue in the SQuAD experiments that used a batch size of 99.5% of the data rather than dropping 0.5% and training in full batch on the remaining dataset, as noted in Footnote 3 and detailed in Appendix E.

  We have re-run this setting with a fix and find the same conclusions, as shown in Figure 23.

- **Dropout as a possible confounder - No changes**

  Our experiments increasing the batch size to Full batch drive the noise to a minimum, but do not remove it entirely due to the use of Dropout in the transformer models. We rerun those models after disabling dropout to verify that the same trends hold.

  While the models can achieve better training error without dropout, the trend that Sign Descent closes most of the gap between SGD and Adam in full batch is preserved.  The results replicating the bottom rows of Figure 5 (momentum), Figure 8 (without momentum) are shown in Appendix D.2, Figure 21. The grid-search validation results are available in Figure 22.

- **Filling missing grid - Improved SignDescent performance on SQuAD**

  We have found that some runs in the grid search had failed due to uncaught OutOfMemory errors. We reran them. The only result affected is the step-size selected for SignDescent with Momentum on Squad, shown in Figure 4 and 5.

  The reviewers did notice that SignDescent was not closing the gap between SGD and Adam on this problem. This issue was partially due to an incomplete grid search. With the initially planned grid-search budget, SignDescent with momentum outperforms Adam on Squad in Figure 5.

- **Added one batch size for normalized runs**

  We have added one batch-size setting in the figures on batch size scaling for normalized methods (normalized GD and sign-descent). Figures 4 and 7 now show results for batch sizes `M, L, XL (new), Full` and their grid-search results (Fig 15-19) have been updated.

---

### Decision · Program_Chairs · 2023-01-20

**Decision:**

Accept: poster

**Justification For Why Not Higher Score:**

No explanations or new hypothesis that can explain the gap between the optimizers.

**Justification For Why Not Lower Score:**

Interesting set of experiments trying to understand the gap between SGD and Adam

**Metareview: Summary, Strengths And Weaknesses:**

This paper is on borderline. All the reviewers felt the paper considered a challenging problem of figuring out the gap between SGD and Adam optimizers and did detailed experiments to challenge the heavy tail noise hypothesis. However reviewers felt the claims of the paper are over the top and not well supported by the current results, 1- Experiments suggest other factors beside heavy tail noise for gap between the optimizers, they don't completely rule out this hypothesis, 2- SignSGD results are not consistent with Adam at all batch sizes. Authors indeed agree and promised to tone down their claims. I am suggesting acceptance assuming authors will honor their response and tone down their claims in the final version.

Pros -
1. The paper presents detailed set of experiments comparing SGD and Adam for larger batch sizes to separate the mini batch noise effects.
2. Results show other factors at play besides the noise in the difference between SGD and Adam

Cons -
1. Over the top claims that are not supported by the experiments.
2. Unclear results around relation between SignSGD and Adam

**Note From Pc:**

if the above contains the word "oral" or "spotlight" please see: "oral" presentation means -> notable-top-5% and "spotlight" means -> notable-top-25%. As stated in our emails, we are disassociating presentation type from AC recommendations

**Summary Of Ac-Reviewer Meeting:**

Reviewers main concerns were 1) the claims in the paper that were not well supported by the experiments, 2) the second part of the paper doesn't have a clear result around SignSGD. As authors promised to tone them down during the response stage reviewers were happy to suggest acceptance.